# Visual segmentation of complex naturalistic structures in an infant eye-tracking search task

**Karola Schlegelmilch** *, **Annie E. Wertz**

Max Planck Research Group Naturalistic Social Cognition, Max Planck Institute for Human Development, Berlin, Germany

* schlegelmilch@mpib-berlin.mpg.de

## Abstract

An infant's everyday visual environment is composed of a complex array of entities, some of which are well integrated into their surroundings. Although infants are already sensitive to some categories in their first year of life, it is not clear which visual information supports their detection of meaningful elements within naturalistic scenes. Here we investigated the impact of image characteristics on 8-month-olds' search performance using a gaze contingent eye-tracking search task. Infants had to detect a target patch on a background image. The stimuli consisted of images taken from three categories: vegetation, non-living natural elements (e.g., stones), and manmade artifacts, for which we also assessed target background differences in lower- and higher-level visual properties. Our results showed that larger target-background differences in the statistical properties scaling invariance and entropy, and also stimulus backgrounds including low pictorial depth, predicted better detection performance. Furthermore, category membership only affected search performance if supported by luminance contrast. Data from an adult comparison group also indicated that infants' search performance relied more on lower-order visual properties than adults. Taken together, these results suggest that infants use a combination of property- and category-related information to parse complex visual stimuli.

**Data Availability Statement:** All files are available from the osf database in the project:

## Introduction

During their first year of life, human infants explore their visual environment in an increasingly selective manner [1, 2], relying on an increasing array of visual properties and category information [3, 4]. Visual abilities develop early and rapidly (e.g., [5]), although some development continues into adolescence (e.g., [6–8]). By about six months of age, basic low-level visual capabilities have emerged which enable infants to distinguish visual pattern within their environment [9–11]. These include grating acuity (i.e., the finest stripes of varying size which can be resolved; e.g., [12]), contrast-sensitivity at higher spatial frequencies (i.e., more narrow changes between light and dark regions; [13–15]), and orientation [10, 16, 17]. These basic functions become more detailed and refined during infancy and early childhood [8, 18–20].

They also provide the basis for higher level visual abilities including the integration of contour segments [21], and the perception of fine detail (i.e., letter acuity; e.g., [22]), all of which support the organization of visual scenes.

Higher order visual competencies, such as visual categorization, also have their onset within the first year of life (e.g., [23–25]; for a review, see [26]) Infants' attentional deployment is modulated by some categorical distinctions by three months of age (e.g., [23, 24, 27, 28]) and they already show sensitivities to particular naturalistic stimuli in the first year of life, including faces [29, 30] and signals of ancestrally recurrent threats like snakes, spiders, and potentially toxic plants [31–33]). This rapid development occurs in the context of varied and cluttered visual scenes consisting of diverse textures, colors, and lighting gradients. However, the previous literature has typically employed stimuli in which such entities are presented in isolation or in well-delineated ways that are not typical of everyday environments. Frequently, entities are well integrated into their surroundings, like books scattered across a child's colorful carpet, or fallen leaves on the playground's sand. The detection of entities within such a scene relies on the ability to perceptually organize its visual information, making additional visual abilities necessary than those necessary for categorization (for a discussion of this problem see [34]). To date, there have been few studies investigating infants' responses to images of naturalistic environments. These studies point to two key factors that affect infants' processing of such scenes: These were (i) sensitivity to naturally appearing visual regularities (i.e., statistically assessed vs. manipulated visual properties; [35, 36]), and (ii) sensitivity to entities with ecological significance (e.g., faces, natural scenes; [37–39]). The aim of the current study is to investigate infants' sensitivity to naturalistic visual information by assessing the effect of different kinds of information (i.e., visual properties, category membership) on visual search performance in a gaze-continent eye-tracking search task.

## Visual regularities affecting segmentation of real-world scenes

Similar to adults, infants are likely to orient towards locations of a visual scene that stick out from their surrounding due to cues of low-level salience (e.g., high contrasts in lighting or color; [37, 40, 41]). Next to this sensitivity to low-level visual information, contour integration (e.g., [42]), and texture segregation (i.e., the effortless segregation of texture patches, for a review see: [43]) are seen as a major mechanisms determining the successful visual organization and identification of scene elements [34, 44, 45]. Scene segmentation is necessary to detect a target in a real-world scene and can be seen as a very basic level of categorization. However, the structures of a naturalistic scene differ in various ways, and their segregation might still pose difficulties for immature visual abilities [46]. Indeed, infants' gaze within their first year of life is still strongly affected by differences in luminance compared to higher-order information such as discontinuities in orientation when viewing artificial stimuli [47] or photographs (e.g., [38, 48]). Still, infants' ongoing gathering of visual information makes it likely that they also base their visual responses on abstract or complex visual properties of the environment [49]. Infants' attention to objects or visual patterns is strongly guided by visual information that provides learning opportunities [50, 51]. Attention to the environment's countless opportunities to receive, organize, differentiate, and accumulate visual information supports the development of visual functions, which are in return adaptive to the visual tasks provided by the environment [22, 29]. This claim is supported by Balas and colleagues [35, 36], who showed that by 9 months of age, infants are sensitive to contrasts between the appearances of naturalistic textures and their statistical transformations. Moreover, infants are surprisingly proficient at processing depth and surface properties [11, 52, 53] suggesting that infants are able to integrate pictorial depth cues into tasks like perceptual organization and action

planning [54]. Further, naturalistic entities within a scene, such as natural and human made entities, differ in their visual properties [55–57], and the presence of certain types of entities or general categories with significance to humans may also affect scene segmentation (e.g., [58]).

## The significance of category information

Research with adults provides many examples of facilitated processing of characteristics that can be encountered in natural environments. For example, the physiology of the visual system responds efficiently to particular aspects of the distribution of spatial frequencies (i.e., changes between light and dark image regions of different amplitudes) in natural environments (e.g., [59–61]). Faster visual processing of, and increased visual memory for, images depicting natural environments compared to images with manipulated properties or non-natural environments (e.g., [62]) suggests ways in which the human visual system adapted to significant aspects of the environment over evolutionary time (e.g., [63]). For example, adults are very fast at detecting animals compared to non-living objects (e.g., [64, 65]). Moreover, information such as social signals (e.g., human faces, [37, 66]), or signals indicating threat [67] are detected fast and in a privileged way.

Importantly, sensitivity to some of these entities and signals is also already evident during infancy. These include categorical distinctions which refer to the animate-inanimate distinction [68–70], and responses to threat signals from spiders or snakes [31, 32, 71]. Moreover, infants showed distinct behavioral responses (i.e., avoidance, increased social information seeking, enhanced learning) when they were confronted with plants compared to other entities [33, 72, 73]. Vegetation plays an important role for humans such that while it provides both food and raw materials, it can be hazardous (e.g., toxins, thorns; [74]). For this reason, the categorization of plants for subsistence strategies was an integral part of ancestral human life [74–76].

These examples indicate that infants' reactions to visual stimuli in their first year are increasingly driven by the ecological or cultural relevance of a stimulus' category [74, 77], suggesting that their visual organization of scenes is also determined by content-related aspects. Nevertheless, it is not yet understood how early sensitivities to certain naturalistic categories relate to the visual properties of these categories. So far, there are only few studies which used real-world scenes to investigate infants' ability to distinguish significant content-related visual information. They frequently included target stimuli with well defined visual characteristics (e.g., scenes with face-targets: [37–39]). The detection of such iconic cues might rely on other mechanisms (e.g., [32]) than the ability to distinguish between the heterogeneous appearances of particular naturalistic categories.

## The current investigation

Here we investigate infants' sensitivity to particular categories and their visual properties using stimuli representing extracts of real-world human environments. The visual structures depicted on the stimuli differed such that they belong to categories with distinct roles over human evolution: vegetation, non-living natural elements (e.g., rocks, water surfaces), and human-made artifacts. These categories have accompanied humans over different timeframes (e.g., categories of the natural environment vs. the younger category of artifacts; (e.g., [78, 79]), and pose visual tasks with different significance on humans (e.g., vegetation as a substantial source of food [80] vs. non-living natural elements such as stones or water (e.g., their role as landmarks during navigation, [81, 82]).

We tested infants' ability to distinguish visual structures by comparing the predictive values of different image attributes on their visual search performance. To do this, we conducted a

visual search task with 8-month-olds including images of real-world structures to investigate the effect of visual properties and categories on scene segmentation (i.e., the detection of a target structure on a background structure). In contrast to stimuli frequently used in studies of categorization and visual development (i.e., faces, objects, or graphics; for overviews see e.g., [11, 24]; but: [35, 36]), we included photographs depicting homogeneous assemblies of natural entities and artifacts. Such visual structures characterize an important proportion of the human environment and their inherent visual properties are relevant for categorization in adults [56, 57].

We used photographs of our three chose superordinate categories: vegetation, non-living natural elements, and artifacts. We chose these categories because they cover important aspects of human environments that are of ecological and social significance, and have been so over evolutionary time. They differ in the role they played during human evolution, the tasks they imposed to humans, and their presence in modern living spaces. For instance, these three categories are part of either a natural or a manmade world (e.g., [55, 83, 84]), they determine the quality and behavioral affordances of a surrounding (e.g., [85–87]), and they can provide organic or mineral material, represent tools, or provide food (e.g., [74, 79]). Moreover, infants typically have visual contact with a variety of instances of each of these categories which provide learning opportunities for some of their aspects.

We used an eye-tracking visual search task in which infants had to find a patch of an image presented on a discrepant background image. This task allowed us to test whether aspects of the images (i.e., visual properties, category membership) affect detection. Infants received a reward (i.e., a colorful butterfly and sound) when their gaze landed on the visual target patch. The reward was included to stimulate visual search. By the age of eight months, infants are able to perform eye movements in order to trigger a reward [88], and gaze-contingent rewards motivate infants' search in eye-tracking experiments if there is no clear pop-out effect for the target [89, 90].

We assessed visual properties selected from research on adult visual categorization of naturalistic entities (e.g., [56, 91, 92]). The current selection of visual properties was chosen based on their potential to discriminate between the general categories of the images used in the current and a previous study of Schlegelmilch and Wertz [93]. The selected visual properties include basic statistical properties that were assessed computationally from pixel greyscale values of the stimulus images, and higher-order characteristics that were assessed by adult raters. Statistical and rated properties are both integral parts of visual categorization in adults (e.g., [94]). Statistical properties differentiate low-level characteristics between general categories and surface properties (e.g., [95, 96]), whereas rated properties are based on perceptual judgments and experience, and are processed in higher visual areas [97]. To date, it is not known what role they play for infants when scenes are visually segmented as a first step in categorization. The code is available on https://osf.io/uyg76/?view_only= 0b1446f6b6504b7193b58fae3a8cb7a3.

In addition to these more structure-related visual properties, we computed a measure of low-level target salience, namely target-background differences in luminance. Luminance contrasts strongly predict infants' gaze (e.g., [37]), so we included luminance differences as a control variable in the analysis. Furthermore, we included variables that quantified the perceived dissimilarity of the images assessed with preschool children and adults in the previous study [93]. Importantly, that study showed that dissimilarity judgments were affected by lower- and higher-level characteristics of the image-structures, and to some extent by the individuals' assumptions about their category membership. Thus, including these dissimilarity judgments allowed us to investigate whether infants' ability to distinguish real-world structures is related to higher-order perceptual judgments in older age groups.

**Table 1. Definitions of the visual properties.**

| Name | Definition | Relevance |
|------|-----------|-----------|
| | Similarity judgments [a] | |
| Perceived dissimilarity | Judgments of visual similarity between the images included in target-background image combinations, assessed in sorting tasks with 4–5-year-olds and adults [93]. Transformed to dissimilarity values. | Subjective judgments of the similarity between the images made by preschoolers and adults rely on perception and assumed category membership of the depicted entities [93]. Related to higher-level image characteristics. |
| | Computational [b] | |
| Luminance | Mean pixel luminance. | The overall lightness or luminance of a structure. Differences in luminance. |
| Alpha | Steepness of the distribution of energy across spatial frequencies (SF) (1/f$^{alpha}$), referring to the proportion of larger changes to more narrow changes between light and dark image regions. | In natural scenes, alpha values are found to lie in a typical range. The adult visual cortex is tuned to these typical ranges of alpha (e.g., [62]). |
| Deviation | Deviation (i.e., area under the curve) of an image's actual SF distribution from the line fitted to this distribution defined by *Alpha* [98]. Deviation distinguishes images in which some SF dominate from images with more evenly distributed SF. | Deviation differs between artifacts, plants, and natural scenes [98]. In naturalistic scenes, low values of deviation relate to higher scaling-invariance (e.g., [99]), in that movement towards the scene does not change its SF-distribution. |
| Entropy | Shannon entropy of pixel luminance values [100]. | Measure of magnitude and predictability of informational content and differentiation. Low values of entropy refer to only a few shades of grey, whereas high values include more differentiated shades. |
| Skew | Skew of the pixel luminance histogram. | Related to impressions of shading and lighting [101,102]. |
| | Rated [c] | |
| Curvature | *Angular vs. curved.* | Perceived curvature supports classification between animate and inanimate objects [92,103]. |
| Regularity | *Regular vs. chaotic.* | Important characteristic for texture and surface discrimination [91,104]. |
| Symmetry | *Symmetrical vs. asymmetrical.* | Symmetry attracts attention in natural scenes (e.g., [105]). Characterizes organic or living things [92]. |
| Depth | *Plane vs. three-dimensional.* | Indicates spatial arrangement of scene elements. Significant for scene segmentation and action planning (e.g., [5]) |

[a] Assessed in sorting tasks with 4–5-year-olds and adults [93].

[b] Computational properties were assessed with functions implemented in Matlab (version R2017b) or provided by literature on image processing [106]. The code is available at https://osf.io/uyg76/?view_only=0b1446f6b6504b7193b58fae3a8cb7a3.

[c] Rated properties were formulated as opposites and judged on a continuous scale by adult participants.

Taken together, the variables we expected to predict infants' search performance belonged to three groups: (a) content-related visual information (i.e., category-congruency and dissimilarity judgments), (b) structure-related visual properties (statistical and rated), (c) low-level salience (i.e., differences in luminance; see Table 1 for descriptions of the variables).

We expected that the selected visual properties could influence infants' search performance in three non-exclusive ways: a) properties of a background image might hinder the detection of the target if they attract infants' attention, b) a property that exceeds the visual abilities of the infant can cause the infant to look-away from the stimulus due to self-regulatory attentional processes [107, 108]; c) stronger differences between a visual property in the target patch and the background image might increase detectability of the target.

Our predictions for the current study were that infants would detect a target patch more easily (a) if it depicts a category which is distinct from the background category, rather than belonging to the same category, (b) if the target's visual properties differ more strongly from the background properties. In addition, infants' search performance can be better interpreted in comparison to adults performing the same task [109–111]. If different patterns of significant predictors occur in adults than in infants, these differences can be related to the infants' visual

and cognitive abilities. Thus, we added data of an adult comparison group to the analysis who had subsequently performed the same experiment.

## Methods

The Ethic Commity of the Max-Planck-Institut für Bildungsforschung has approved the study i2018-06.1, former MatSoC 2018/03 by written consent.

### Participants

The final infant sample was $N$ = 39 eight-month-olds (age: $M$ = 8 months, 11 days; range = 8 months, 0 days to 8 months, 29 days; 18 female), recruited from urban and suburban regions of a large European city. We chose 8-month-olds for the current investigation given their successful performance on gaze-contingent search tasks in previous studies [88, 90, 112], and early evidence for distinctions between general categories within the second half of the first year of life [25]. An additional two infants were tested but excluded because no data could be assessed due to problems with the eye-tracker. All infant participants had normal vision.

In response to a helpful comment from a reviewer, we decided to include a comparison dataset of adults ($N$ = 20, age: $M$ = 32 years; range = 23 to 58 years; 12 female) who performed the identical experiment as the infants. All adults had normal or corrected to normal vision by wearing contact lenses.

Participants were recruited from our internal participant database and tested in the Max Planck Institute for Human Development, Berlin, Germany. Parents gave written consent for their child's participation. Participation was compensated with 10 Euros and infants additionally received a participation certificate.

### Stimuli

The 27 images which comprised the search stimuli of the current study were selected from a set of 60 greyscale images used in the study of Schlegelmilch and Wertz [93] that investigated the impact of visual properties on categorization in preschool children and adults. The images depict extracts of real-world structures representing one of the three superordinate categories of either vegetation (e.g., foliage, bark, grass), non-living natural elements (e.g., water surfaces, rocks), or artifacts (e.g., cloth, office supplies; the 27 images used in the study are shown in Fig 1A). Each entity occupies the full size of the image. They were photographed by the first author, or downloaded from license-free online image repositories (pixabay.com, pxhere.com, gettyimages.de).

The stimuli for the search task each consisted of one background image into which a circular patch of a different image was inserted as target. The size of the background image measured 1280 × 1024 pixels, leading to 32° × 25.5° of visual angle during presentation (*vis*), the target patch measured 235 pixel (6° *vis*) in diameter. Targets were placed at one of 10 possible locations arranged in a circle with ca. 710 pixel (18° *vis*) distance to the screen center (Fig 2). Each target had a blurred border to prevent that the circular contour of the target was used as cue. With the same intention, a pattern of blurred circles along the outer contours of the 10 possible target locations was included in each background image. In order to obtain stimuli with moderate target saliency, we applied a salience algorithm to all possible target-background combinations and locations, using the statistical software Matlab (version R2017b, http://www.mathworks.com). We chose the Graph-Based Visual Saliency algorithm (GBVS; [113]) specified for discontinuities in luminance and orientation. GBVS had reflected infant gaze patterns well (for a discussion of salience applied in infant research see: [41]). We then chose target-background combinations in which a target was quantified as at least moderately

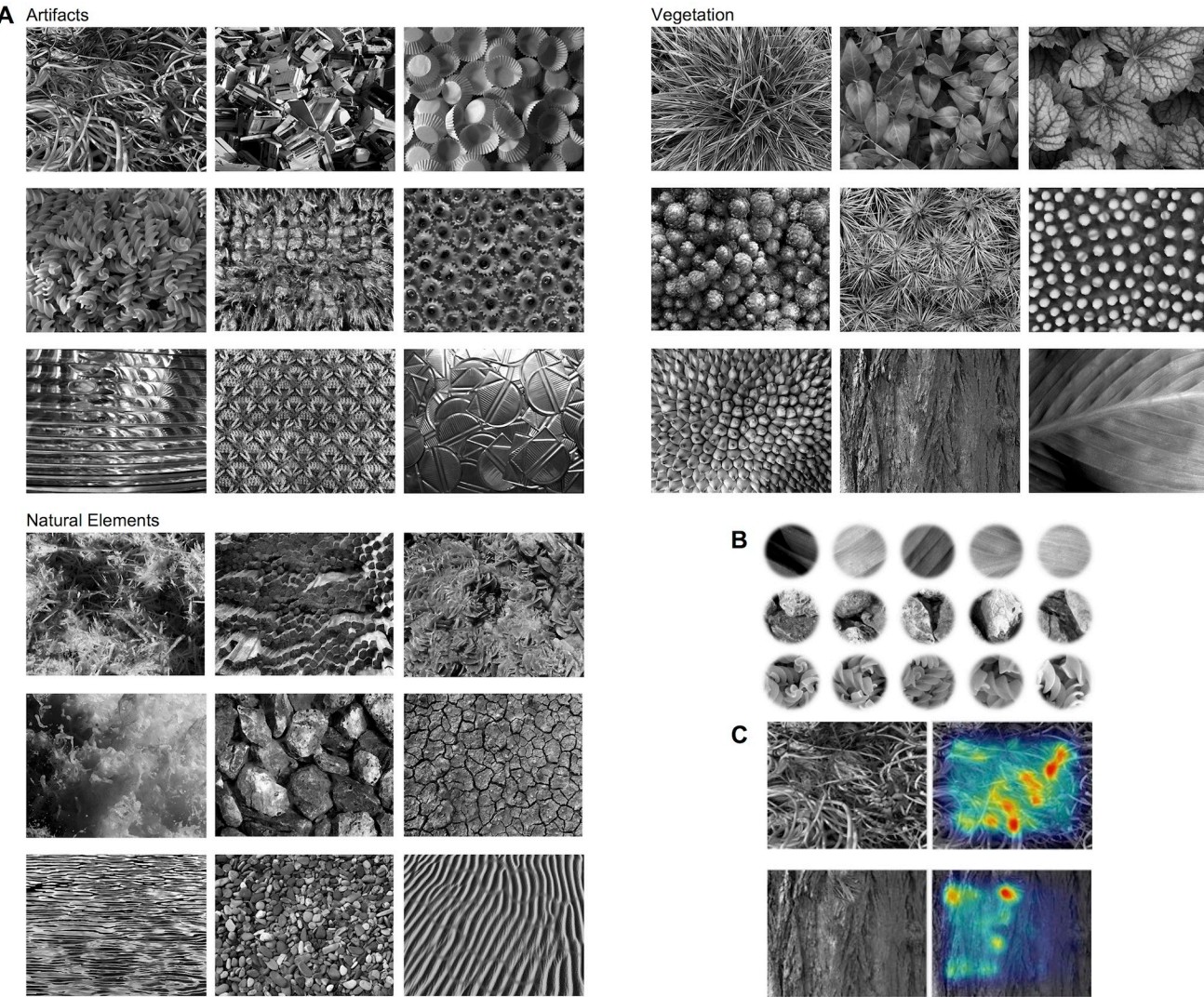

**Fig 1. Creation of the search stimuli used in the study.** (A) The 27 images used in this study, grouped by category, in the format in which they were used as backgrounds in the search task. Within each category, the images are arranged left to right according to decreasing values in rated depth. (B) Examples of the five target patches sampled from each of the 27 images. Targets could appear at one of 10 possible target locations, see Fig 2. (C) Stimuli with moderate target salience were created by placing different target samples at each of the 10 locations, respectively, and comparing the results with a salience algorithm (GBVS; [113]). Targets that were salient (indicated in the two examples by orange to red overlay) without being the only salient region of the stimulus at the respective location were selected. The 261 stimuli identified through this process were converted to three monochromatic colors (green, blue, red) each. From this set, we produced the eight versions of the experiment with 36 test trials each following these restrictions: The stimuli (i) were balanced over target and background categories, (ii) were balanced over the crossed factors category- and depth-congruency, and (iii) no image should appear more than twice as either target or background in any of the eight versions. Taken together, the eight versions of the full experiment included 288 test-trials. The resulting frequencies of the defining factors in these trials are reported in S1 Table in S1 Text.

salient, but not as the only salient region of the stimulus (Fig 1C). Note that the GBVS salience map differs from the computation of the property diff_luminance in that the salience map uses various factors to predict gaze, whereas diff_luminance solely assesses the effect of target luminance on the overall variability of luminance within the image.

Target patches and background images either depicted the same or a different category, leading to stimuli with congruent categories (e.g., vegetation target on vegetation background), or incongruent categories (e.g., artifact target on vegetation background). In the previous

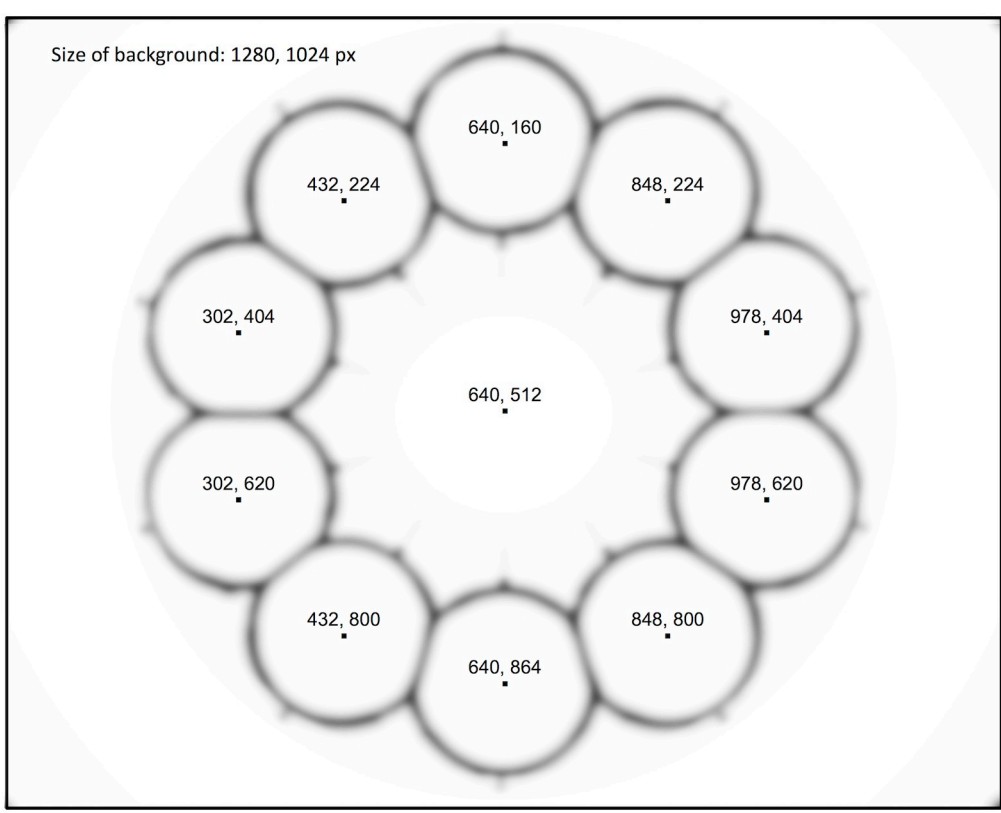

**Fig 2. The arrangement of target locations and blurred contours.** Central pixel coordinates of the ten possible target locations, in equal distance to the background center (first value: horizontal from top left, second value vertical). The contours around all of the possible target locations—arranged as circles in a ring—were blurred on each background image, and the target patch was included in one of the circles, respectively.

study with preschool children and adults, depth cues were an important predictor of categorization decisions. In order to prevent depth from being a confound in the analysis of category-congruency, we balanced images with high and low rated depth within the respective categories. We then crossed category-congruency (congruent vs. incongruent) with a control variable we termed depth-congruency (similarly high ratings of depth vs. contrasting ratings of depth in the target and the background image). In addition to its role as control variable, depth-congruency contains higher order information about the target-background relationship in depth, and thus complements the property depth, which refers only to the background image.

The computationally-assessed properties were included in our analysis as difference variables: We first partitioned the background images without the target patch as well as the background image including the target patch into squares (size = 256 px by 256 px) which fitted the size of the target patch. Next, we calculated a property's variance a) between the partitions of the background, and b) between the partitions of the background including the target patch. Finally, we subtracted the background's variance from the target-plus-background's variance. This procedure was applied to the 261 stimuli covering all target-background image combinations and their target locations included in the study. The obtained difference variables represented the impact of a target property on the variability of this property in the whole stimulus (termed diff_*property-name*). High values of difference variables were obtained if a target exhibited very high or very low levels of a property, which exceeded the range of the background-image's property-levels in either direction. Low values of difference variables resulted

from backgrounds in which the levels of a property varied between high and low extremes, so that the property level of the target could not substantially increase the background's variance. If an infant's detection performance was predicted by a difference variable, the infant must have been sensitive to discontinuities of this property—either within the background image (background difficulty) or between background and target (detection facilitation). Note that the impact of a difference variable does not provide information about an infant's respective sensitivity for particular high or low levels of the property. In contrast, the visual properties based on human ratings had been assessed for entire images [93], so these ratings could not validly represent the small regions of the images used as target patch. Consequently, we only included the background's rated properties in our analysis without assessing target-background differences. To make the experiment more engaging for infants, we used three alternating monochromatic colors for the search stimuli. This was done by transforming the greyscale images to HSL color space with the hues: 90˚ (green), 210˚ (blue), 330˚ (red), using the software Adobe Photoshop (Adobe Photoshop CC, Version 2017.0.0). The target and background always shared the same color within a stimulus. Increasing infants' attention generally reduces movement, increases the periods of recorded gaze and leads to better eye-tracking data quality [114].

In sum, the target-background combinations of 27 images on 10 possible locations and presented in three different colors led to 261 different stimuli that crossed category congruency and pictorial depth congruency. The frequencies of the factors defining the image combinations are provided in S1 Table in S1 Text, their respective visual properties are accessible on https://osf.io/uyg76/?view_only=0b1446f6b6504b7193b58fae3a8cb7a3.

## Experimental design and procedure

First, a target sticker was placed on the infant's forehead and the infant was seated in a dimmed room in front of the eye-tracker (EyeLink 1000 Plus; SR Research Ltd. 2013–2015) either in a baby chair ($N = 37$) with the caregiver right behind, or on their caregiver's lap ($N = 2$). A welcome video was played during the set-up of the eye-tracking camera (EyeLink 1000 Plus High-speed Camera with a 16 mm lens), which was placed approximately 60 cm in front of the target sticker as recommended by the manufacturer [115]. Monocular pupil and corneal reflections were assessed in a sampling rate of 500 Hz. The presentation monitor (50" display, with 1280 by 1024 pixel resolution, and 400Hz CMR refresh rate) was set at a distance of 140 cm away from the infants' eyes to approximately fit the trackable area of 32˚ *vis* by 26˚ *vis* in accordance with the manufacturer's suggestion. After set-up, the experimenter stepped behind a curtain from where the infant and caregiver could be monitored on a video screen and started the experiment.

At the beginning of the experiment and after at least each eighth trial, five-point calibrations were conducted with calibration targets alternating in color, form and sound. Changes in these characteristics increase infants' interest in the calibration procedure, leading to higher data quality [114]. The trial started when the average calibration error was below 1˚ *vis* (see Fig 3).

The first five trials of the experiment were practice trials. They started with an easy-to-detect target patch on a simple background and gradually increased in difficulty. Then, 36 test trials were presented in randomized order. In each trial, the color of the stimulus was randomly altered (green, blue, and red). Additional easy-to-detect practice trials were initiated if it seemed that the infant became unaware of the task after several misses without receiving a reward. If the infant became inattentive, showed fatigue, or if the caregiver requested a break, we paused the experiment for a few minutes, or terminated the experiment prematurely.

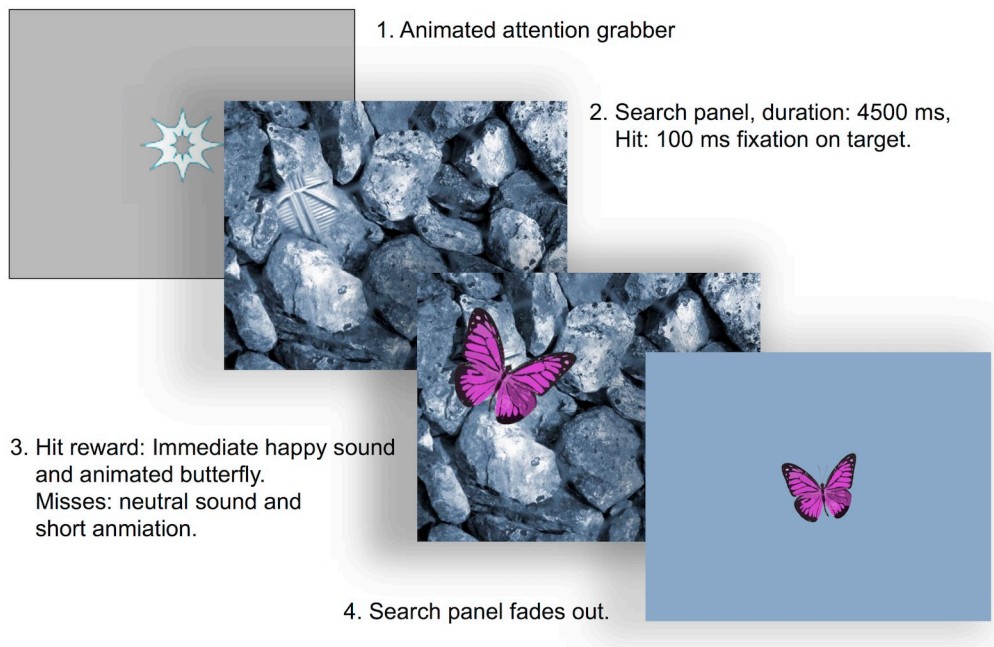

**Fig 3. Trial example.** This example shows one trial with a monochromatic blue search stimulus, and one of the five butterflies that were alternated as rewards. Each trial included a central attention grabber of 5° vis in diameter. As soon as the infant's gaze rested on its central area (2.5° vis in diameter) for 100 ms, one of the search stimuli was shown for a maximum of 4500 ms. If the infant's gaze rested on the target patch before timeout for at least 100 ms, rewarding music started to play, then a colorful butterfly loomed out of the target's center and moved to the center of the screen, and the trial ended. If a target was missed, the butterfly was only shown for a shorter time, accompanied by a neutral sound. Directly after the butterflies disappeared, a new trial started. Every fifth trial, an attention grabber was shown at a peripheral location in addition to the central location. If the infant's gaze was not recorded within the central region of the attention grabber—for example because of inattentiveness or changes in the distance between the eyes and the eye-tracking camera—an additional calibration was initiated and the camera set-up was corrected if necessary. The length of the full trial varied due to when infants fixated on the attention grabber, and whether they detected the target.

There were eight versions of the experiment that alternated between participants. Each version included a different selection of 36 target-background image combinations taken from our 261 stimulus variants. No target or background image was included more than twice in one version. To avoid memory effects when an image was repeated, its second occurrence was part of a different target-background image combination and used a different target location and color.

## Results

### Preliminary analysis and data reduction

Infants completed a median of 34 trials (range = 26 to 36 trials), with a median of 88% of gaze recorded by the eye-tracker per trial (range: 1% to 100%). In the following analysis, we included only trials in which infants attended to the stimulus long enough to have the opportunity to detect it. Thus, we decided on distinct criteria for hit and miss trials. This had the advantage of reducing noise in the latency analysis (e.g., [116]), while respecting infants' attention spans during unsuccessful search in a realistic way. Using an identical proportional inclusion criterion in the hit and miss trials would have unnecessarily excluded trials in which infants may have searched for the target for a considerable amount of time but nevertheless failed to detect it within the 4500 ms trial. We therefore defined our criteria as follows: Trials

in which infants detected a target (hit) were accepted if they had a minimum of 80% of recorded gaze. Trials in which a target was not detected (miss) were accepted if they included at least 1240 ms of recorded gaze, which was the median of the hit latency for the whole sample (for studies defining minimum periods of recorded gaze see e.g., [37, 41]). Applying these inclusion criteria led to $Mdn$ = 32 valid trials per infant (range = 23 to 36).

Adults completed 1439 trials (range = 35 to 36). The inclusion criteria that we applied in the adult sample was identical to that of the infants, and led to 1429 valid trials. Due to recruitment issues during the COVID-19 pandemic, we recruited 20 adult participants who performed two runs of different experiment versions each. This led to $N$ = 40 runs and approximately equals the number of all runs presented to the infant sample ($N$ = 39).

Infants detected a target in $Md$ = 39% of the trials, range = 17% to 55%. In contrast, adults detected the target in almost all the trials ($Md$ = 100%, range = 92% to 100%, equating to only 27 misses and 1402 hits). The adults' ceiling effect was the result of using a trial duration that was originally adjusted to infants. Because of the small number of misses, we only analyzed detection latency for the adult data. Table 2 shows gaze and performance characteristics of infants and adults. Longer fixation durations, later initiation of first saccades, and shorter total durations of gaze spend on the scene are common for infants compared to adults (reviewed in: [117]). We will report the results of the adults together with the infants' results for each of the respective models. The data of the infant and adult samples is publicly available at (https://osf. io/uyg76/?view_only=0b1446f6b6504b7193b58fae3a8cb7a3).

## Statistical analysis of search performance

In order to investigate the effect of image characteristics on search performance, we assessed the binarily coded dependent variable (DV) success (hit, miss) and the continuous DV latency, which represented the time until a target was detected if it was a hit. These two DVs covered infants' reactions to aspects of stimulus salience, detection difficulty and background complexity. To account for individual differences, non-normality and unbalanced conditions which are common in infant and eye-tracking data [118, 119], we conducted mixed effect models with the R-package lme4 [120]. For the generalized linear effect models (GLMM) on the DV success, we used the function glmer and specified a binomial error structure. The units of analysis included as random effects on success were participant, background image, and target location. On the DV latency, linear mixed effect models (LMM) were conducted with the function lmer. For latency, the random effects participant and background image were defined,

**Table 2. Gaze and performance characteristics of infants and adults.**

|  | Infants | | | Adults | | |
|---|---|---|---|---|---|---|
|  | *Md* | *SD* | *range* | *Md* | *SD* | *range* |
| Hit-rate[a] | .39 | .1 | [.17, .55] | .98 | .02 | [.92, 1] |
| Latency until hit (ms)[a] | 1471 | 353 | [991, 2127] | 632 | 106 | [238, 4152] |
| N fixations until hit[ab] | 8 | 3.1 | [2, 16] | 2.5 | 1.1 | [1, 9.2] |
| Rate 1st hit-fixation[b] | .09 | .06 | [0, .26] | .26 | .18 | [0, .69] |
| Proportion of recorded gaze[c] | .84 | .8 | [.6, .97] | 1 | .02 | [.89, 1] |
| Mean fixation duration (ms)[ab] | 467 | 100 | [276,717] | 230 | 25 | [197,311] |

*Note*. Infants' data is averaged by id ($N$ = 39), adults' data by id and run ($N$ = 40).

[a] Assessed from data included in the analysis, with recorded gaze $\geq$ .8 in hits, $\geq$ 1240ms in misses.

[b] Fixations > 50 ms are included.

[c] Proportion of registered gaze points within trials in raw data.

whereas target location did not improve the model fit and was not included ($\chi^2(1) = 0.04$, *n.s.*). Residual and specification diagnostics were carried out with the R package DHARMa [121] and by inspection of residual plots. Influential cases were diagnosed with regard to DFBetas (function influence; R-package lme4 [120]). The significance of predictors was assessed by comparing the current model with a model reduced by the respective predictor in chi-square likelihood-ratio tests (LRT) with the R-function Anova (package car; [122]).

To avoid problems of interdependencies between IVs (see e.g., [123]), we reduced the number of IVs in each comparison by conducting separate models for different research questions (e.g., the impact of computationally assessed visual properties). For these models, we estimated the effect of collinearity by Variance Inflation Factors (VIF; [124]) with the function vif (R-package car; [122]) and only combined IVs in models if VIF values remained below 2.5.

**Effects of covariates.** We tested if stimulus color [green, red, blue] generally affected search performance. The factor color [red, green, blue] did not predict infants' search performance (success: $\chi^2(2) = 1.6$, *n.s.*; latency: $\chi^2(2) = 3.7$, *n.s.*), nor performance of adults (all $\chi^2(2) < 1.4$, *n.s.*), confirming that the colors we chose to enhance infants' interest in the study did not lead to differences in the detectability of the target, see also Section B in S1 Text.

Because movement during remote-mode eye-tracking substantially affects data quality [114, 125], we calculated the covariate movement as the maximum of absolute change in head-camera distance within fixations during each presentation of a search stimulus (for details see Section A in S1 Text). Movement was included as a covariate in all models.

In our analysis, we were interested in the impact of the predictor variables beyond target-background differences in luminance. We therefore included diff_luminance as fixed effect in all models. We also included the interaction term between diff_luminance and the other predictor variables if it significantly improved the model compared to a model with fixed effects only, as assessed in a LRT (R function anova, package stats; [126]). Significant interactions between diff_luminance and another predictor variable indicate that the other predictor's impact on target detection is affected by the targets' high luminance contrast.

The two runs of the experiment executed by the adult comparison sample included different stimuli. Increasing experience with the task might have led to better detection performance in the second run. We therefore conducted an analysis of the effect of the factor run [1, 2] together with the covariates diff_luminance and movement on adults' detection latency. This showed that run did not significantly contribute to the model fit ($\chi^2(1) = 0.8$, $p = .378$), whereas the covariates diff_luminance ($\chi^2(1) = 7.1$, $p = .008$) and movement ($\chi^2(1) = 135$, $p < .001$) both affected latency. This confirms that performance between the two runs did not differ in the adult data. Yet, in order to rule out any practice-effect, we nevertheless included the factor run as additional covariate in the adult models.

## The impact of content-related visual information on detection performance

**Category-congruency.** Here, we investigated whether differences between the background category and the target category affected search performance. Category-congruency and the control variable depth-congruency were included in the models as predictors.

Infants' GLMM on success was improved if it included the interaction terms between category- and depth-congruency and diff_luminance compared to the same model with only main effects, $\Delta\chi^2(1) = 8.4$, $p = .004$. LRTs on success indicated significant contributions of the fixed effect category-congruency ($\chi^2(1) = 7.3$, $p = .007$) and the interaction term between category-congruency and luminance ($\chi^2(1) = 10.5$, $p = .001$). Congruent categories led to a higher probability to detect a target if combined with greater diff_luminance. Incongruent categories led

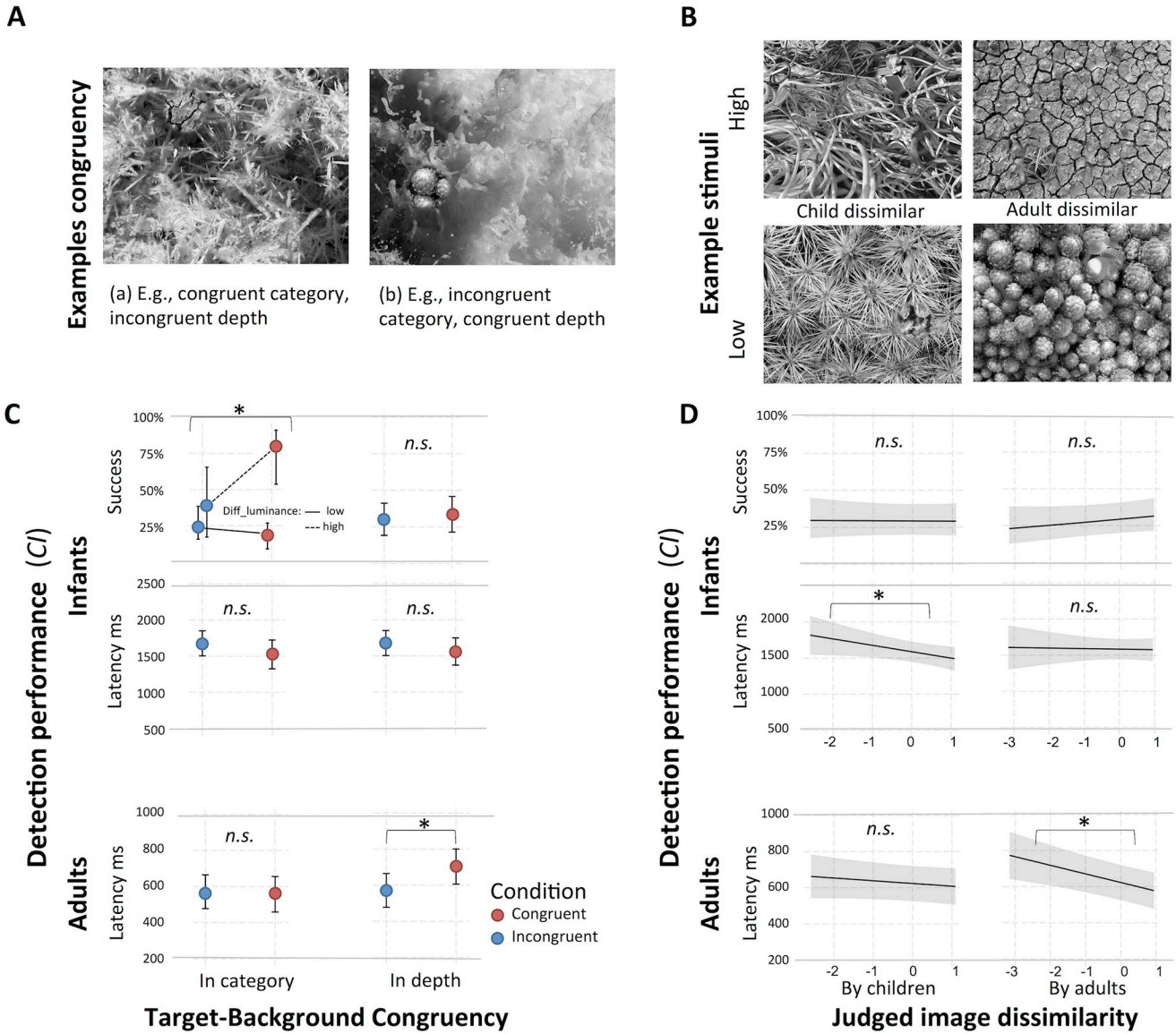

**Fig 4. Infants' and adults' performance as functions of target-background congruency and image dissimilarity.** Stimuli examples (A, B) and marginal effects of the models (C, D) conducted with independent variables relating to categories and higher-level image characteristics. (A) Category-congruency and depth-congruency of target and background images. Two out of the four possible combinations are shown in greyscale. The stimuli alternated between three monochromatic colors during presentation, see Stimuli paragraph in Methods section. (B) Stimuli including target and background images with low or high dissimilarity, judged by preschool children and adults in a previous study [93]. (C) Top: Interactions between diff_luminance and the congruency variables, diff_luminance low = 5%, high = 95% percentile. High diff_luminance is related to higher detection success for target categories that were congruent with the background categories (red), compared to incongruent categories (blue), whereas category congruency does not affect performance if differences in luminance are low. Depth congruency was not significantly affected by diff_luminance.

to better detection performance than congruent categories if the full range of diff_luminance is taken into account, and they were affected less by differences in luminance. Therefore, incongruent categories differed more strongly from congruent categories when combined with greater than with lower differences in luminance, see top row in Fig 4C.

In the LMM on latency, category-congruency did not contribute to the model, $\chi^2(1) = 3.6$, $p = .059$. The control variable depth-congruency neither improved the model fit on success

(main effect: $\chi^2(1) = 0$, $p = .960$; interaction: $\chi^2(1) = 3.3$, $p = .069$), nor on latency, $\chi^2(1) = 2$, $p = .160$, Fig 3 and Table 3.

The adults' LMM on latency was not improved by the inclusion of an interaction term between diff_luminance and the congruency variables, $\Delta\chi^2(2) = 4.9$, $p = .084$. LRTs indicated a significant contribution of depth-congruency ($\chi^2(1) = 21.2$, $p < .001$), whereas category-congruency did not contribute to the model ($\chi^2(1) = 0.2$, $p = .646$), Fig 4C, Table 3.

**High-level image dissimilarity.** With the continuous variables child_dissimilarity and adult_dissimilarity, which were taken from ratings of the target and background images in a previous study [93], we examined whether detection success was influenced by the higher-level similarity judgments of older age groups.

In the infants' GLMM on success, neither child_dissimilarity nor adult_dissimilarity improved the model fit, as indicated by the LRT (both $\chi^2(1) < 1$, *n.s.*). However, infants' detection latency was predicted by child_dissimilarity ($\chi^2(1) = 4.2$, $p = .042$), but not by adult_dissimilarity, indicating that infants were faster at detecting targets if they were perceived as more dissimilar by preschoolers, but not by adults, Fig 4D top rows.

In adults, the variable adult_dissimilarity improved the model fit on latency ($\chi^2(1) = 11.8$, $p < .001$), whereas child_dissimilarity did not contribute to the model ($\chi^2(1) = 1.2$, $p = .281$), Fig 4 and Table 3. Adults were faster at detecting the target if adults in another study [93] had judged it as more dissimilar to the background, whereas preschoolers' judgments from that study did not affect adults' detection.

**Table 3. Results for congruency and image dissimilarity.**

| Property | GLMM on success | | | | LMM on latency | | | |
|---|---|---|---|---|---|---|---|---|
| | Log-Odds | 95% CI | z | p [a] | b (ms) | 95% CI | t | p [a] |
| | Infants | | | | | | | |
| | Category-congruency | | | | | | | |
| Diff_luminance | 0.22 | [-0.22, 0.65] | 0.99 | .322 | -88 | [−193, 17] | -1.64 | .103 |
| Category-congruency | **-0.61** | **[-1.05, -0.17]** | **-2.70** | **.007** | -157 | [−320, 6] | -1.89 | .059 |
| Depth-congruency | 0.01 | [-0.47, 0.45] | -0.05 | .960 | -130 | [−310, 51] | -1.41 | .159 |
| Diff_luminance: Category-congruency | **0.83** | **[0.33, 1.32]** | **3.25** | **.001** | | | | |
| Diff_luminance: Depth-congruency | 0.49 | [-.04, 1.02] | 1.82 | .069 | | | | |
| | Image dissimilarity | | | | | | | |
| Diff_luminance | **0.71** | **[0.44, 0.98]** | **5.13** | **< .001** | -90 | [-192, 13] | -1.73 | .085 |
| Child_dissimilarity | 0.01 | [-0.17, 0.16] | -.08 | .941 | **-89** | **[-174, -3]** | **-2.04** | **.042** |
| Adult_dissimilarity | 0.11 | [-0.06, 0.27] | 1.28 | .199 | -7 | [-91, 76] | -0.17 | .863 |
| | Adults | | | | | | | |
| | Category-congruency | | | | | | | |
| Diff_luminance | - - | - - | - - | - - | **-.51** | **[-95, -6]** | **-2.23** | **.026** |
| Category-congruency | - - | - - | - - | - - | -12 | [-63, 39] | -0.46 | .646 |
| Depth-congruency | - - | - - | - - | - - | **137** | **[155, 217]** | **11.82** | **< .001** |
| | Image dissimilarity | | | | | | | |
| Diff_luminance | - - | - - | - - | - - | **-62** | **[-107, -17]** | **-2.68** | **.007** |
| Child_dissimilarity | - - | - - | - - | - - | -15 | [-43, 12] | -1.08 | .281 |
| Adult_dissimilarity | - - | - - | - - | - - | **-48** | **[-75, -20]** | **-3.44** | **< .001** |

[a] *P*-values obtained by chi-square likelihood-ratio tests.

## The effect of visual properties on detection performance

Target-background differences in computational properties (i.e., diff_deviation, diff_alpha, diff_entropy, and diff_ skew) and rated background properties (i.e., curvature, depth, regularity and symmetry) were analyzed in separate models. None of the models included interaction terms with diff_luminance. This led to four analyses conducted to assess the impact of visual properties on infants' search performance.

In the infant GLMM of computational properties on success, diff_luminance contributed to the model fit with $\chi^2(1) = 19.8$, $p < .001$. Of the structure-related predictors, only diff_deviation contributed with ($\chi^2(1) = 22.2$, $p < .001$), in that higher values of both variables lead to a higher probability to detect the target. Latency was predicted by the structure-related property diff_entropy, which contributed to the fit of the LMM with $\chi^2(1) = 8.5$, $p = .004$. Stronger target-background differences in diff_entropy led to a faster detection of the targets, see Fig 5 and Table 4.

In the infant GLMM on success including the rated background properties, diff_luminance contributed to the model with $\chi^2(1) = 27.9$, $p < .001$ and depth affected the model fit with ($\chi^2(1) = 4$, $p = .046$), in that higher values of diff_luminance, but lower values of background depth, lead to a higher probability that a target was detected. Depth also contributed to the fit of the LMM on latency ($\chi^2(1) = 11.1$, $p < .001$), with higher values of depth leading to longer detection latencies.

LRTs indicated that no other visual properties affected infants' detection performance, see Table 4 for all results and Fig 5 for stimuli examples of significant properties.

Adults' latency was predicted by the structure-related property diff_alpha in the LMM including computational properties ($\chi^2(1) = 5$, $p = .024$) such that stronger target-background differences in alpha led to a faster target detection. In the LMM on latency with rated properties, the control variable diff_luminance was the only variable that contributed to the model fit ($\chi^2(1) = 4.1$, $p = .043$), see Fig 5B and Table 4.

## Did infants detect the targets by coincidence?

In order to investigate whether infants may have fixated on the targets by coincidence, we compared the number of fixations during the presentation of the search-stimulus on each of the 10 possible target locations without a target to the number of fixations on the target. We considered fixations to the 10 locations on which targets could occur to be areas of interest (AOIs) and not all fixations because infants might have learned that targets are located a certain distance from the screen center. A GLMM specified for count data on the numbers of fixations with the predictor location (the target contrasted to the 10 AOIs) and participant as random effect indicated that there were less fixations to any AOI without a target than to the target itself, LRT on the IV location ($\chi^2(10) = 599$, $p < .001$), all contrasts $p < .005$.

We then compared the number of first fixations on the target to the mean number of first fixations on any of the 9 AOIs without the target. First fixations to targets occurred more frequently ($Md = 3$, range = 0–8) than first fixations to non-target AOIs on average ($Md = .89$, range = .22–2.56), as confirmed by a t-test ($t(44) = 6.81$, $CI(95\%) = [-2.8, -1.5]$, $p < .001$). The proportion of first fixations on the target, relative to the number of first fixations to the 10 possible target locations is shown in Fig 6, in addition to the chance-level of hitting any specific one of the 10 locations ($p = .1$, red line). Only three of the 39 infants fell below chance-level. These results confirm that infants' target detection was non-accidental.

## Discussion

Here we investigated which image characteristics (i.e., category, luminance, structure-related visual property) affected 8-month-olds' ability to detect a discrepant image patch on a complex

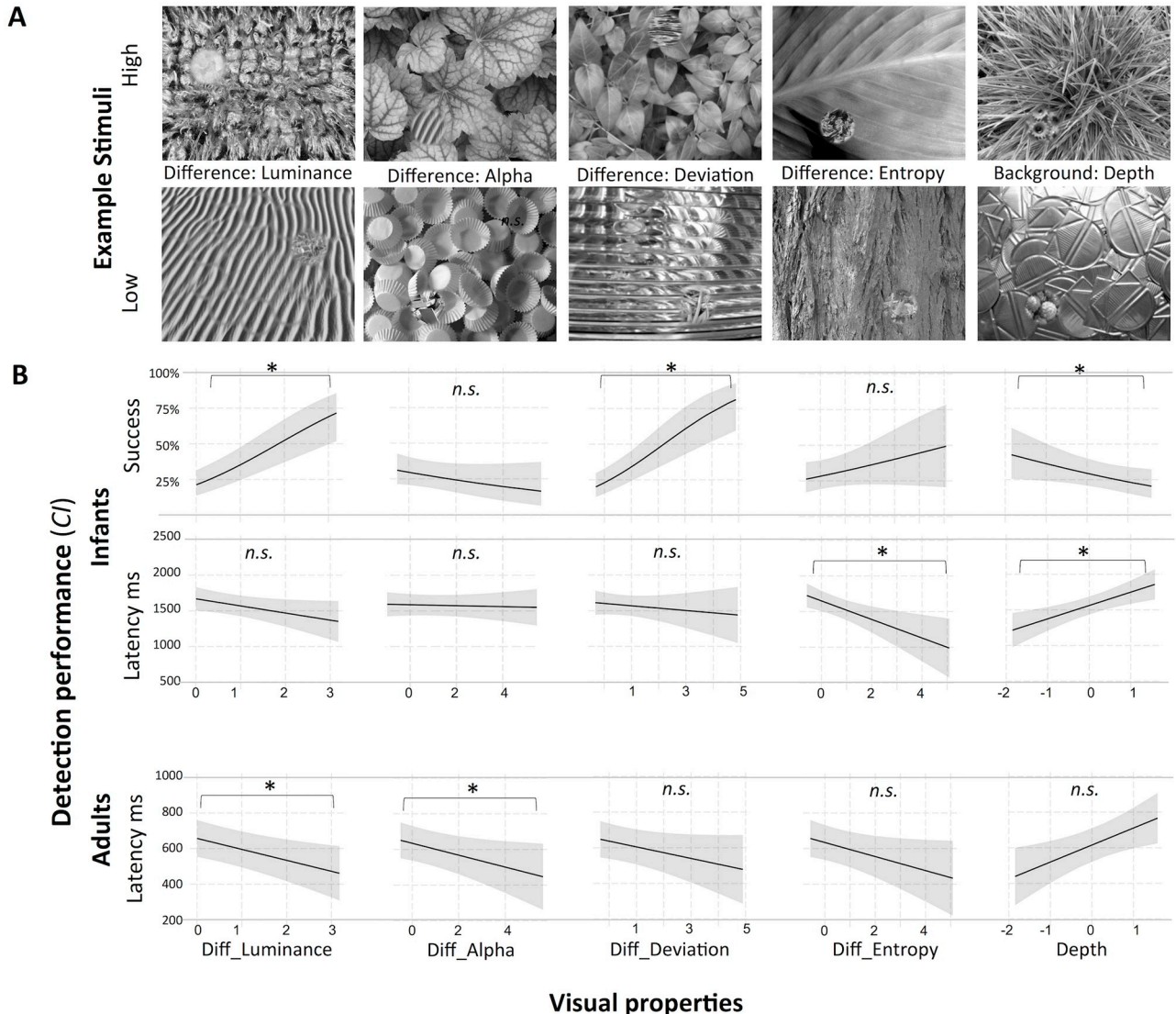

**Fig 5. Structure-related visual properties and their effect on search performance.** (A) Examples of search stimuli with the respective low and high values of visual properties that contributed significantly to any of the models. (B) Visual properties as functions of search performance and participant group, estimated as marginal effects in the models. Asterisks indicate significant contributions of the variable to the full models, see Table 4. Diff_luminance is estimated as single variable together with the covariates of the respective model groups (i.e., infants: movement; adults: movement, run).

background image using a gaze-contingent eye-tracking search task. The images depicted one of the three superordinate categories: vegetation, non-living natural elements or artifacts.

Our results indicate that infants attended to combinations of higher- and lower-level visual image characteristics to distinguish complex naturalistic patterns. Consistent with the previous literature (e.g., [37]), detection performance was affected to a large extent by differences in luminance. However, going beyond the results of previous studies, we found that structure-related visual properties of the images, such as deviation, entropy, and rated depth, predicted detection performance independently of luminance. Furthermore, judgments of image-dissimilarity by preschoolers—but not by adults—predicted infants' detection performance. Yet,

**Table 4. Results of structure-related properties that predicted detection performance.**

| Property | GLMM on success | | | | LMM on latency | | | |
|---|---|---|---|---|---|---|---|---|
| | Log-Odds | 95% CI | z | p [a] | b (ms) | 95% CI | t | p [a] |
| | Infants | | | | | | | |
| | Computational target-background difference [b] | | | | | | | |
| Diff_luminance | **0.62** | **[0.35, 0.90]** | **4.45** | **< .001** | -42 | [-148, 65] | -0.76 | .446 |
| Diff_alpha | -0.14 | [-0.33, 0.05] | -1.4 | .160 | -2 | [-93, 90] | -0.04 | .968 |
| Diff_deviation | **0.55** | **[0.32, 0.78]** | **4.71** | **< .001** | -37 | [-124, 49] | -0.84 | .402 |
| Diff_entropy | 0.19 | [-0.06, 0.45] | 1.5 | .134 | **-128** | **[-214, -42]** | **-2.92** | **.004** |
| Diff_skew | 0.09 | [-0.1, 0.28] | 0.92 | .356 | -42 | [-118, 34] | -1.08 | .283 |
| | Rated background property | | | | | | | |
| Diff_luminance | **0.70** | **[0.44, 0.96]** | **5.28** | **< .001** | -95 | [-189, -1] | -1.97 | .054 |
| Curvature | 0.07 | [-0.2, 0.34] | 0.51 | .611 | -11 | [-117, 95] | -0.2 | .844 |
| Depth | **-0.31** | **[-0.61, -0.01]** | **-2** | **.046** | **187** | **[77, 297]** | **3.38** | **.004** |
| Regularity | 0.19 | [-0.22, 0.6] | 0.9 | .370 | -66 | [-220, 88] | -0.84 | .412 |
| Symmetry | 0.11 | [-0.28, 0.5] | 0.54 | .590 | 128 | [-20, 276] | 1.7 | .106 |
| | Adults | | | | | | | |
| | Computational target-background difference [b] | | | | | | | |
| Diff_luminance | -- | -- | -- | -- | -32 | [-78, 14] | -1.36 | .175 |
| Diff_alpha | -- | -- | -- | -- | -31 | [-62, 0] | -1.98 | .048 |
| Diff_deviation | -- | -- | -- | -- | -16 | [-54, 22] | -0.84 | .403 |
| Diff_entropy | -- | -- | -- | -- | -25 | [-65, 16] | -1.2 | .231 |
| Diff_skew | -- | -- | -- | -- | -22 | [-54, 11] | -1.34 | .187 |
| | Rated background property | | | | | | | |
| Diff_luminance | -- | -- | -- | -- | -44 | [-88, -1] | -2.02 | .044 |
| Curvature | -- | -- | -- | -- | -33 | [-97, 31] | -1.02 | .320 |
| Depth | -- | -- | -- | -- | **55** | **[-16, 125]** | **1.52** | **.143** |
| Regularity | -- | -- | -- | -- | -37 | [-133, 60] | -0.74 | .46 |
| Symmetry | -- | -- | -- | -- | -48 | [-140, 43] | -1.04 | .312 |

*Note.* Visual properties were included together with the covariate movement as fixed effects.

[a] *P*-values obtained by chi-square likelihood-ratio tests.

[b] Assessed as difference between the properties' variance within the background image alone and within the background including the target patch, see Stimuli in Method section.

the impact of categorical information on infants' detection performance depended on the stimulus' target-background difference in luminance.

In the current study, targets were detected non-accidentally, indicating that infants learned to search for the targets and were able to direct their attention to discontinuities in the appearance of the structures. The current findings differ from earlier eye-tracking search tasks with infants (e.g., [39, 89]) in that targets did not represent a delineated object. Instead, photographs of complex naturalistic surfaces or assemblies of elements alternated as targets and backgrounds. A target was only defined by it being a discrepant structure patch to the background and by leading to a visual reward if looked at. Therefore, our results are relevant for research on image segmentation and visual search (e.g., [49, 127, 128]).

## Structure-related visual properties affected detection performance

Target-background differences in deviation and entropy explained a similar or even higher amount of variance in detection performance than differences in luminance ($R^2_{marginal} = .042$

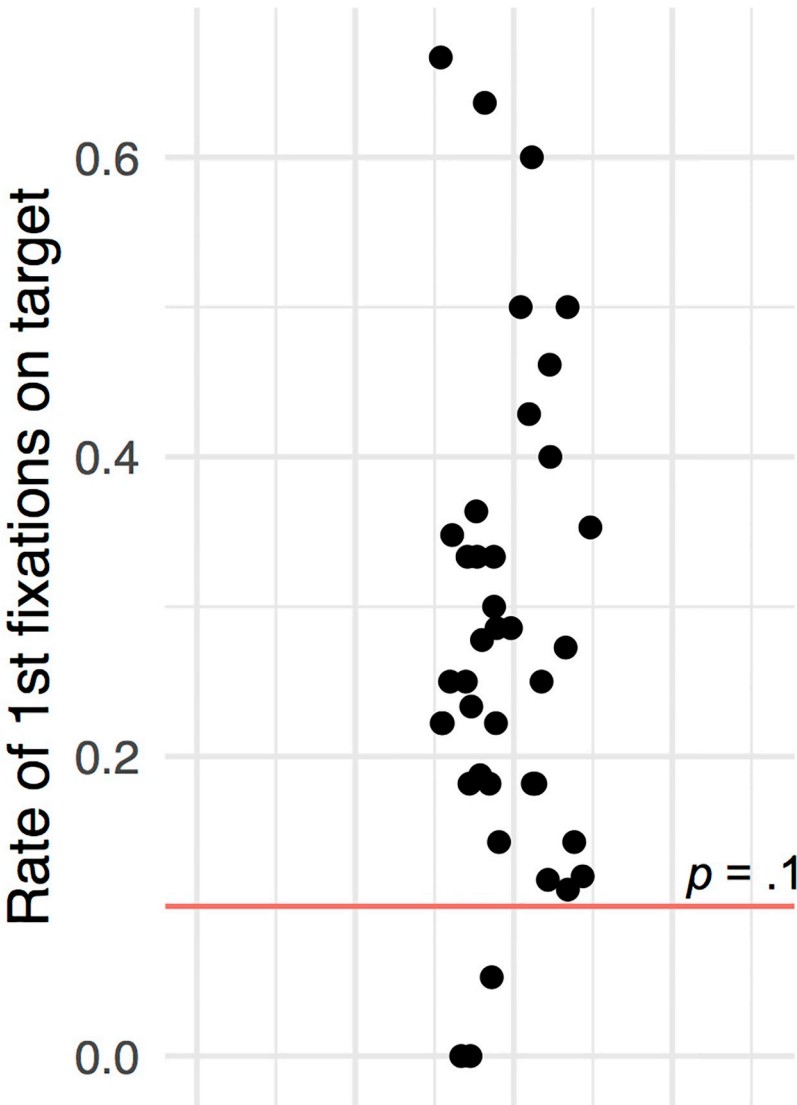

**Fig 6. First fixations on areas of interest including the target.** Points indicate the rate of an infant's number of first fixations on targets relative to the number of the first fixations on any of the 10 possible target locations (AOIs), independent of their including the target or not (rates: Md = .25, sd = .16, range = 0 to .67). The red line indicates the chance-level of a first fixation on a particular target location, if only fixations within the AOIs are considered. Points are scattered sideways to avoid overlap.

for diff_deviation vs. $R^2_{marginal}$ = .019 for diff_luminance on success; $R^2_{marginal}$ = .033 for diff_entropy vs. $R^2_{marginal}$ = .003 for diff_luminance on latency; R-function *r.squaredGLMM*, package MuMIn; [129]). This is intriguing because infants' attention at this age is strongly affected by luminance (e.g., [37]) and several aspects of these visual properties may have facilitated infants' detection of the discrepant target structure.

It is possible that differences in the amount of shades of grey, and in the amount of spatial scales that vary in the properties entropy and deviation, affected infants' detection

performance. A structure defined by high deviation values is dominated by only some spatial frequency scales, providing similarly shaded regions of repetitive sizes, whereas a structure defined by low deviation includes all spatial frequency scales. Structures with high values of entropy include similar numbers of each of the 256 possible shades of grey of an 8 bit image, while in low entropy high proportions of only some shades lead to less differentiated shading or more monotonous structure regions. Accordingly, greater differences in deviation and entropy between the target and background were associated with more fine-grained, cluttered patterns and differentiated contrasts, versus less detailed, smoother, or repetitive image regions. It is interesting that infants were sensitive to these image properties because infants' immature processing of fine detail and lower sensitivity to contrast may lead to uncertainty and make it difficult for them to detect variability in visual structures that differ in these ways. Sensitivity to uncertain visual information might nevertheless be beneficial for the infant for various reasons. It can lead to strategic behavioral reactions such as further exploration, avoidance, or social referencing (e.g., [27, 77]) and support basic distinctions of significant categories. Sensitivity to uncertain visual information might also underlie infants' novelty preference [29] and in young children's choice of actions that resolve the greatest amount of uncertainty [130–132]. For example, if infants move towards a visual structure with low deviation, the distribution of small to large spatial pattern will remain the same due to its scaling invariance. It is plausible that infants become sensitive to such phenomena without being fully able to process every visual detail. Sensitivities to these phenomena are particularly advantageous, since differences in scaling invariance distinguish important general categories (e.g., artifacts vs. natural elements; [98]). Perhaps due to these adaptive and explorative behaviors, infants are sensitive to visual properties that varied in their amount of informational uncertainty and informational complexity, and attended to discrepancies within these properties in the current task. This would explain how greater differences in target-background deviation and entropy led infants to detect the discrepant target patch.

However, one may then ask why greater differences in other computational visual properties did not affect infants' target detection performance. For example, alpha represents statistical aspects of naturalistic scenes, and different values of alpha have even been found to affect adults' processing speed, recognition, and visual memory (e.g., [62, 98, 133, 134]). And indeed, adults' detection latencies in the current study were shorter if targets differed more strongly from the background in alpha in the current search task. Nevertheless, in the current study, alpha did not predict infants' detection performance (Table 4). Similarly, certain ranges in alpha did not predict judgments made by younger children, but did so for older children and adults [46]. It is possible that variations in alpha are much more difficult to distinguish for young children, because alpha defines *proportions* of spatial scales, but not the *range* of the spatial scales included in the structure. The discrimination of variations in alpha makes sensitivity to the full spectrum of spatial scales necessary [46] and can be seen as higher-order statistic compared to deviation.

## Depth cues, but not shape predicted detection performance

We found that high rated depth of the background image hindered infants' target detection performance. Infants start to be sensitive to stereoscopic depth and to pictorial depth within a few months after birth [11, 135, 136]. Nevertheless, the dark regions and high contrast contours which are typical for shading-defined pictorial depth might have diverted infants' gaze and complicated the detection of the target. Adults' detection performance was not hindered significantly by higher background depth. Instead, adults took advantage from depth-congruency, in contrast to infants whose performance was not affected by depth-congruency. Recall

that this variable stated if the level of rated depth in the target image was the same as in the background image. In contrast to background depth, the visual processing of depth-congruency requires comparisons between pictorial depth cues within image regions, which may have been beyond infants' visual abilities. The reasons why depth affected target detection may therefore be different in infants and adults. A photographic representation of complex three-dimensional arrangements might challenge an infant's perceptual abilities, and could potentially lead to either disengagement from the task or the search for further information [3, 137]. Thus, an alternative explanation is possible: Infants did not disengage from the depth cues because spatial characteristics of scene elements or their arrangement provided opportunities for further visual exploration (e.g., [138]) and significant attentional learning processes [3, 4] that were more rewarding than searching for the target.

Interestingly, none of the other rated properties—which were all related to shape characteristics or their arrangement within the background image—affected target detection performance in infants or adults. One might argue that higher ratings of these properties were as interesting or demanding for the infants as the lower ratings. Yet, this explanation is unlikely, because (i) curved shape is preferred to angular shape very early in life [29], (ii) symmetry is processed in a basic way by 1-year-olds and may attract attention because it provides learning opportunities [139], and (ii) repeated elements which represent regularity are reliably used as backgrounds in infant search paradigms (e.g., [89]). As with pictorial depth, these findings were obtained with graphical stimuli and therefore may not fully describe the effect of these properties on infants' visual abilities if they are present in complex naturalistic scenes.

The current results show that depth cues can play an important role in scene perception and segmentation. However, it remains unclear why the particular shape properties defining complex naturalistic structures we assessed in this study did not affect detection performance in either of the participant groups.

## Infants' visual search relates to preschoolers', but not adults' similarity judgments

Remarkably, judgments of image dissimilarity made by preschoolers, but not adults, in a recent study [93] predicted infants' search performance. This raises the question of which aspects of structure perception are shared between infants and preschoolers, but are less relevant for adults.

The perception of complex naturalistic structures is a hierarchical organizational process and relies on several perceptual mechanisms: smaller elements can be grouped, compared, segregated, or perceived in their configural relations, leading to more global elements with which the operations can be repeated [140, 141]. Some visual abilities that are important for structure perception—such as spatial acuity, contrast sensitivity, and particularly the higher-order ability of perceptual integration—are only adult-like after the preschool years [6, 8, 15].

In the current search task, infants' detection performance was less affected by variables that relied on higher-order processing, such as alpha or depth congruency, whereas these properties enhanced performance in the adult participants. Likewise, during the sorting task from which the dissimilarity judgments were derived, preschoolers more reliably attended to properties that required little hierarchical processing, while adults integrated properties of several hierarchical processing levels into their judgments [93]. Neuroimaging studies have found an increase of horizontal intra- and interhemispheric connectivity at least until 13 years of age, as well as increasing feedback connectivity from extrastriate visual areas to V1—changes which are thought to be involved in the detection, grouping and spatial integration of distributed visual elements [15, 142, 143]. It is therefore likely that immaturities in perceptual integration

that contribute to difficulties processing complex visual images are an important overlap between infants and preschoolers.

Another overlap could be that preschoolers' perceptual judgments relied on their experiences with the structures which are likely related to different aspects of the entities than those of adults. For example, young children's relation to their environment is to a large extent guided by explorative actions. These are described as the primary behavioral mechanism for generating perceptual information [144–146]. In contrast, adults actions more strongly refer to the context or usage of the entities (e.g., [79]). Moreover, perception of infants and young children before school age is less affected by cultural norms than perception in older children and adults [147]. Such fundamental differences in the gathering of experiences very likely affect perception in young children differently than that of adults. Next to related immaturities in visual abilities, overlaps in a learning- and exploration-driven relation to the environment between preschoolers and infants might therefore have caused the predictive value of preschoolers', but not adults' judgments on infants' performance. With this, the finding provides an example for age-dependent variations in the significance of visual aspects of the environment.

## Did categorical information affect infants' detection performance?

One of our main interests was whether certain superordinate categories that have significance for humans—vegetation, artifacts, and natural elements—affect scene-segmentation in infancy. However, we did not find clear evidence that differences in these categories between target and background images affected target detection or segmentation ability in infants or adults. Infants' search performance was only affected by category information that was supported by differences in luminance. Detection probability of a target was higher if it belonged to a category which was *congruent* to the background category and also differed more strongly from the background in luminance. In contrast, *incongruent* category combinations were less affected by differences in luminance, but led to better detection success than congruent category combinations if the full range of luminance differences is considered, Fig 4C. In adults, luminance differences did not interact with categorical information, and there were no differences in detection latency between congruent and incongruent category combinations.

It is difficult to separate visual properties from category information. Accordingly, we do not see categories as independent or opposite from visual properties—categories must necessarily be defined by a set of properties. In order to rule out biases in the stimuli that might have led to a facilitated detection of congruent or incongruent categories in infants, we analyzed if any of the selected visual properties differed between category-congruent and category-incongruent stimuli. Table 5 compares the visual properties of congruent and incongruent category combinations, respectively, within the stimuli of all trials that were presented to the participants in the eight versions of the experiment. The comparison shows that none of the visual properties included in the current analysis differed between the congruent and incongruent category combinations.

A similar supporting role of luminance differences for infants' detection of categorical information was found in previous studies investigating the effect of low-level salience on the detection of faces. In those studies, stronger luminance contrasts within the face target facilitated its detection in infants younger than 1 year of age ([39], but see: [37]) and stronger luminance contrasts between competing stimuli hindered the detection of the faces in 4-month-old infants [41]. Fig 4C shows that—compared to category-incongruent stimuli—category-congruent stimuli profited much more from high luminance contrasts, whereas low luminance contrasts only resulted in a minor disadvantage. Thus, the current findings do not provide

**Table 5. Visual properties as function of category-congruency.**

| Variable | M congruent [a] | M incongruent [a] | d | t [b] | p [b] |
|---|---|---|---|---|---|
| Diff_luminance | 0.70 | 0.61 | -0.11 | 0.89 | .373 |
| Diff_alpha | 0.50 | 0.42 | -0.90 | 0.74 | .458 |
| Diff_deviation | 0.47 | 0.50 | 0.03 | -0.25 | .802 |
| Diff_entropy | 0.48 | 0.35 | -0.15 | 1.16 | .248 |
| Diff_skew | 0.09 | 0.18 | 0.09 | -0.77 | .444 |
| Curvature | -0.01 | -0.08 | -0.08 | 0.60 | .546 |
| Depth | 0.27 | 0.31 | 0.04 | -0.30 | .767 |
| Regularity | -0.19 | -0.27 | -0.09 | 0.69 | .492 |
| Symmetry | -0.05 | -0.15 | -0.10 | 0.83 | .408 |

[a] Stimuli with congruent ($N = 96$) or incongruent ($N = 192$) target-background category combinations.

[b] T-test between category-congruent and -incongruent stimuli.

*Note*. The analysis refers to $N = 288$ data points distributed over 36 test-trials in the respective 8 versions of the experiment.

evidence for a better distinction of structures belonging to a different general category, as we had expected based on the existing literature on categorization in infants (e.g., [39, 68, 69, 72]).

One reason for this may be because evidence for sensitivity to particular content-related visual information in photographic scenes comes from studies with iconic and object-like targets. In these studies, targets were commonly the only exemplar of their general category in the context of a different general category (e.g., landscapes including an animal, interiors surrounding a person's face or body; [39, 64]). Visual abilities underlying the detection of a particular structure patch (as in the current study) may be somewhat distinct from those underlying the detection of delineated objects. For example, hierarchical perceptual integration and grouping mechanisms play a stronger role in the perceptual organization of structures compared to bounded objects (e.g., [141]).

## Limitations and future questions

It cannot be ruled out that infants used cues to detect the targets beyond those analyzed. For example, they might have learned to associate rewards with round areas of a certain size—despite the efforts to hide the contours of the targets. This might have led them to preferably fixate round salient patches, leading to faster detection if the round patch actually included the target. However, we think that such cues did not strongly alter search performance, since otherwise, rated background curvature would have affected detection significantly, and it did not.

The current visual properties only represent a small selection of many possible visual properties that could play a role in the perception and segmentation of naturalistic structure. It cannot be ruled out that some properties that were not assessed in the current study affected detection performance. Still, we think that the current selection of visual properties highlighted several important aspects of visual development (e.g., action or exploration related attention, differences in processing effort between lower and higher order statistics), and included important characteristics of the environment that affected search performance (e.g., their distribution of spatial scales and contrasts). Future investigations should build on some of the current findings. In particular the effect of an increase in pictorial depth of the background image on infants' detection performance needs to be evaluated more deeply. Teasing apart the roles of attention or explorative reactions to functional spatial information on the one hand, and distractions due to shading and contrast on the other hand, would deepen insight into the impact of functional significance on early vision development.

The monochromatic stimuli we used allowed us to focus on structure-related visual properties. If the stimuli would have maintained their original colors, target detection would have been dominated by color differences. However, the effect of category information on search performance might have been affected by this decision: Color cues provide important information supporting the discrimination of significant categories [56, 148].

With regard to the ambiguous influence of category information on the segmentation of visual scenes, it would be interesting conduct additional investigations with different age groups using a similar search task. Target-background combinations of different superordinate categories could be compared to combinations of sub-groups within particular superordinate categories. By also examining the interplay of luminance contrasts and category contrasts this search task would extend developmental research on visual salience (e.g., [37, 48, 149]).

One key question concerning infants' visual processing of naturalistic stimuli is whether basic visual abilities, which are typically assessed with graphic stimuli, transfer to naturalistic scenes. In adults, visual responses to naturalistic stimuli exceed the performance that can be expected from responses to graphic stimuli [150]. In future studies, it might therefore be useful to add psychophysical search stimuli to the experiment that vary in contrast and fine detail. An assessment of the individual infant's low-level visual abilities would allow to better uncover if sensitivity to content- or structure-related properties during the segmentation of naturalistic scenes correlates with the ability to react to fine detail in abstract stimuli.

## Conclusion

The current study revealed that 8-months-olds were able to search for discontinuities in photographs depicting naturalistic surfaces or assemblies of elements. The task to search for a discrepant target image patch was solely defined by a gaze-contingent reward. Infants detection performance was primarily affected by luminance differences and by structure-related visual properties such as scaling invariance (i.e., deviation) and entropy. Differences in category membership did not affect search performance independent of luminance differences.

The pattern of results suggests that infants' gaze was largely guided by statistical properties. Higher-order visual information such as pictorial depth cues hindered detection. Possibly, opportunities to explore this significant property increased infants' distraction by high pictorial depth. Visual properties affecting infants' detection performance differed from the properties affecting performance of an adult comparison group. For example, in contrast to the adult participants, infants' gaze was not affected by the statistical property alpha or by the level of rated depth between target and background image. Additionally, infants' detection performance was predicted by preschool children's but not adults' perceptual judgments of the same image combinations, whereas adults' detection latency was only predicted by the dissimilarity judgments of adults. Thus, maturing visual abilities necessary for naturalistic structure perception such as the perceptual integration of distributed elements seems to affect the perception of these naturalistic structures in infants.

Infants were sensitive to variations in the amount of grey scales defining a structure, and to variations in spatial frequency distributions indicating scaling invariance (assessed by the property deviation). These properties typically differ between artifact categories and significant domains of natural categories (i.e., vegetation, natural elements [59, 62, 98]). Variations in these visual properties include visual information that exceeds the infants' low-level visual abilities [6, 8, 46]. The sensitivity to variability within these properties we found in infants may be related to the importance of these properties in distinguishing the significant category domains. However, given the many reasons why infants might attend to one visual cue but not another—including exploration and visual learning—the significance of visual information

applies to more than particular entities or category domains. The current study showed that infants' visual abilities allow them to perceptually organize complex structures within their environment by reacting to visual information, even if it was uncertain or incomplete. Thus, visual aspects or physical qualities that are part of an infant's developmental tasks in supporting his or her interaction with the environment may be as important as particular entities or categories.

In congruence with other studies using naturalistic images in controlled laboratory settings (e.g., [36, 38, 39]) we argue that the inclusion of naturalistic images in infant vision research is important and might lead to different results than research with artificial objects or graphic stimuli.

## Supporting information

**S1 Text. Supplementary results.** A. The effect of movement on detection performance. B. Did changes between the differently colored stimuli affect infants' detection performance? S1 Table: Number of trials per factor level, in the original experiment and in the infant and adult data.
(PDF)

## Acknowledgments

We thank our participants and their parents, Janek Stahlberg and the members of the Max Planck Research Group Naturalistic Social Cognition for their assistance. We also thank the team of the Department of Developmental Psychology: Infancy and Childhood at the University of Zurich for the supportive discussions of this project. Over the course of the project, Karola Schlegelmilch was external Fellow of the International Max Planck Research School on the Life Course (LIFE).

## Author Contributions

**Conceptualization:** Karola Schlegelmilch, Annie E. Wertz.

**Funding acquisition:** Annie E. Wertz.

**Investigation:** Karola Schlegelmilch.

**Methodology:** Karola Schlegelmilch.

**Supervision:** Annie E. Wertz.

**Visualization:** Karola Schlegelmilch.

**Writing – original draft:** Karola Schlegelmilch.

**Writing – review & editing:** Annie E. Wertz.

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
