## [Decision Letter · Decision Letter 0]

27 May 2021

PONE-D-21-01158

Visual segmentation of complex naturalistic structures in an infant eye-tracking search task

PLOS ONE

Dear Dr. Schlegelmilch,

Thank you for submitting your manuscript to PLOS ONE. After careful consideration, we feel that it has merit but does not fully meet PLOS ONE’s publication criteria as it currently stands. Therefore, we invite you to submit a revised version of the manuscript that addresses the points raised during the review process.

Two expert reviewers have assessed your work. Both reviewers commend the important and challenging topic you address, and provide generally encouraging comments. Both reviewers also provide detailed and constructive comments aimed at improving the clarity of your manuscript. Reviewer 2 additionally however raises some serious concerns about the soundness of your results and your interpretation. I am unsure at this time whether it will be possible to address these concerns, as this will require you to check and/or modify your analyses, develop further analyses to eliminate alternative interpretations, or readjust the interpretation of your results. Nevertheless, if you are successful your study would be a nice addition to the literature, so I look forward to receiving your revised work.

As a separate note, I would like to apologizes for the unusually long time that the manuscript was under review. Unfortunately, one expert who had originally agreed to review your work unexpectedly dropped out of the review process and caused significant delays. 

We look forward to receiving your revised manuscript.

Kind regards,

Guido Maiello

Academic Editor

PLOS ONE

Journal Requirements:

[Wealso
thank the team of the Departmentof Developmental Psychology: Infancy and Childhoodat theUniversity of Zurichfor their support. ]

 [The author(s) received no specific funding for this work]

Reviewers' comments:

Reviewer's Responses to Questions

**Comments to the Author**

1. Is the manuscript technically sound, and do the data support the conclusions?

Reviewer #1: Partly

Reviewer #2: Partly

2. Has the statistical analysis been performed appropriately and rigorously? 

Reviewer #1: Yes

Reviewer #2: Yes

3. Have the authors made all data underlying the findings in their manuscript fully available?

Reviewer #1: Yes

Reviewer #2: No

4. Is the manuscript presented in an intelligible fashion and written in standard English?

Reviewer #1: Yes

Reviewer #2: Yes

5. Review Comments to the Author

Reviewer #1: There is increasing interest in how we develop sensitivity to natural scene statistics, and how infants learn to segment complex visual scenes and attend to relevent information, and this study has a significant contribution to make to this area of research. The task is a visual search task where infants are presented a target circle on a background - the 'category' that the target is from is either a congruent natural texture, or incongruent. The authors conclude that performance on this task is a result of a combination of perceptual and categorical properties of the stimuli.

The paper represents a huge amout of work, both in image and data analysis and in data collection all of a high quality. There are some areas that should be addressed, either in a response to review or making edits to the manuscript. We have chosen major revisions only because it seems there is some work to be done to make the authors argument clearer, and does not reflect the quality of the study itself. We're looking forward to reading the final published paper in a journal club in the not too distant future. *note, paper reviewed with the assistance of doctoral researcher, hence 'we' throughout!

- Infant perception of stimuli: Depth, Pixel-wise measures, and use of colour stimuli

1. The use of depth congurency is an interesting measure. As the authors note, infants this age are sensitive to some depth cues. However, there is evidence that children don't necessarily combine cues to achieve adult like depth perception. Does this have implications for the relevance of the congruency of depth cues in this study? As the authors themselves note in the discussion, it's hard to disentangle 'depth' from images with higher perceived depth having more high contrast areas which are likely to attract the infant.

Nardini, M., Bedford, R., & Mareschal, D. (2010). Fusion of visual cues is not mandatory in children. Proceedings of the National Academy of Sciences, 107(39), 17041-17046.

2. Many of the measures used by the authors are based on pixel wise measures (e.g. mean luminance calcualted with each pixel), and some of the differences in stimuli levels will have likely been small. Are these realistically discriminable to an infant, and can any of the findings be explained by a limited infant visual system?

3. The study uses monchromatic stimuli in 3 colours R,G, and B to try and encourage infant engagment with the task. This unfortunately may have inadvertantly added noise to the measures the authors are collecting data from. For example, brightness perception varies as a function of hue - two colours of equal luminance do not necessarily appear equally bright (Helmholtz–Kohlrausch effect). Although the authors don't find a main effect of colour on performance, which does reassure somewhat that there's no 'hidden bias' brought in by using colour in this way, the authors should be aware that values calculated on greyscale versions of stimuli may not neccessarily reflect perception of chromatic stimuli.

- Categorisation

3. Categories: the paper leans heavily on the use of the word 'category' when discussing the rationale and findings. The paper states that 'infants attend to combinatons of category AND property related cues to distinguish naturalistic patterns' implying that the paper considers categories to be an entirely separate entity to the lower level property related cues. We think that the main evidence for there being a category effect is most effectively shown in figure 3 - by there being a greater chance of target detection for congruent stimuli when contrast differences are large, and the inverse effect when contrast differences are smaller. Are there alternative explantions that don't call on categories - for example is this is actually just an ability to spot an outlier in a statistical distribution rather than an effect of categories?

- Clarity of paper and general comments

4. The paper makes a lot of effort to be clear - the table of definitions is very good and helpful, and overall, the writing is excellent. However, as a result of there being so much included in the paper, it does mean it is in places confusing, or that the key findings get lost in the paper. We're not advocating that the authors remove sections from the paper, but we do think the paper might benefit from a heavy edit for consicesness. The paper has a huge amount to offer which is currently being lost a little along the way.

5. Should 'intensity' be contrast or luminance throughout? All the variables listed could be measured in 'intensity' so it was a little confusing in places.

6. line 347 - one of these 'incongruents' should be 'congruent' - or alternatively we have misunderstood the way that the model fits to the data.

Reviewer #2: This paper investigates infants’ ability to recognize and discriminate visual patterns by virtue of their category (vegetation, artifact, non-living natural) membership. This was assessed through a visual search task, where a small target patch of an image was embedded in a background image. The target patch was always drawn from a different image, but that image could be from the same category as the background, or from a different category. There are various detours and other considerations, but the overarching hypothesis is that category membership, per se (as opposed to various concomitant low-level visual differences that manifest between images from different categories), would be noted and drive looking toward the target. In general, the work is sound, I really appreciate this area of investigation, and the melding of natural scene image analysis and psychophysics in an infant study. It is a nice niche that would benefit from more work. That said, there were aspects of the study (and the interpretation of results) where I had some concerns.

Overall, the exposition itself, especially around methods and results, sometimes lacked clarity and motivation, and could be more refined and deliberate. I will try to offer some concrete suggestions here.

I had concerns with the data screening. As it stands, the screening is based on behavioral outcomes (throwing away a “hit” because recorded gaze was <80%, but applying a different criteria for miss trials). This seems potentially problematic. I would strongly encourage the authors to apply just one, erring-on-the-side-of-inclusivity, criteria across the board, before any considerations of performance or outcomes.

There were phrases scattered throughout the text that had the feeling of technical terms, but had vague and unclear meaning, such as “physically intense cues”, “perceptual difficulty”, “prominence”, “familiarity”, “level of property”, “property value”, “less [/more] distinct category combinations”, “processing advantages”, “discriminated statistically”, “difficulty of the images”. It would help the exposition if these terms were replaced with more specific, definitive ones, or at least defined/operationalized.

In places, the technical terms themselves could be sharpened. Why not just call “intensity” / “low-level intensity” / “physically intense cues”, simply mean luminance? Why not call “diff_mean” ‘diff_scaleInvariance’? Etc.

Sometimes this can affect understanding of central claims. For instance, I am not clear what is meant by “...visual property could influence infants' search performance in two non-exclusive ways: a) their prominence within a background image might hinder the detection of the target.” Here, it is not clear (to me) what is meant by “prominence” of dimensions that have no natural valence? Could the authors reword and clarify?

I think the authors can make a stronger case for “Were targets detected by coincidence” I would be interested to see other comparisons between the target aoi, and the average of the other 9 aoi’s, e.g.: # fixations until aoi (i.e. target aoi vs. average of other 9 aois), time to aoi, dwell time on aoi, ‘success rate’ (proportion of trials on which target, versus other 9 aois, was reached). These “chance levels” (‘coincidence’) should be reported wherever possible (e.g. Figure 1 and Figure 3) since they give a good frame of reference, at least for Intensity 0 conditions.

I was a little unclear on how many trials and subjects contributed to each ‘data point’ (e.g. target-background combination, or at least categories of target-background). Something about the math was not clear to me (“27 images on 10 possible locations and presented in three different colors led to 260 different stimuli”). More detail could be given about the data itself, and the breakdown by conditions, colors. If my math is right, it works out to be about 5 trials per image, 15 per image if we collapse over color? But, those are divided by 4 if we wanted to, say, just compare performance at intensity 0?

I think figure 1 has an incorrect y axis on the scatter plots (I expect it to be RT in ms) or am I missing something?

I do not understand how the authors are using hits and misses when, typically defined, misses are just 1-hits. Why not just code performance as percent correct?

Apologies if I missed it, but what latency is entered if the target is not found (miss)?

Nearly everything - certainly all the figures - from the “supplementary materials” need to be in the main text. As well, the figures could use more annotation and labels, and more detailed captions.

The statistics wind up being a bit complex due to all the factors and varying tests in different contexts. I think the paper could do with some more data visualization. (As it is, we only have Figure 2, which does not even have data points, and the caption does not say anything about the nature of the fits, etc.). Some of this would be mitigated by my earlier suggestion to move other Figures and information from the Supplement into the main text.

Do the authors have a reference for the alternating-color-stimuli design they used? This seems a rather extreme way to help maintain vigilance. I know the authors found that color did not predict performance, but what of differences between the colors?

Now, turning to the results themselves and their interpretation. For what I think is the most natural, straightforward comparison - the ‘success rate’ of whether infants found target patches differentially, depending on if it had a category match/mismatch with the background - they seem to have a null result. This is captured in Figure 3 at intensity level 0 (intensity refers to mean luminance, which, in levels 1-3, was artificially increased on target patches to facilitate infants’ search). In this comparison, there is no difference between success as a result of target-category membership - the core contrast of the whole study - and actually, target direction rates themselves are so low as to approach chance (i.e. fixating the target ‘accidentally’ as the infant simply scans the scene). Null results are fine of course, but the introduction and discussion (layering in further analyses and speculation) tended to bury the lede here. Do the authors agree? Shouldn’t this finding be more central in the discussion?

Then, why was mean luminance (intensity) varied at all? This manipulation needs a lot more justification and explanation, especially as it winds up being the central driver of the “main” (counterintuitive and unexpected) results, given its interaction with category membership. It is hard to think of any reason to expect luminance to interact with categorization. In fact, I think the default stance would be that putting category information on top of a (much more impactful) cue like relative luminance would tend to either effectively discourage the use of category information (by rendering it largely unnecessary), or, as a practical, ‘signal/noise’ matter, tend to obscure any relatively small effects of category against the backdrop of much larger effects of luminance.

Then, while I am sympathetic to the author’s attempts to link the present results to category formation, I am hoping the authors can make a stronger case (this point is relevant also to my next one below). All the individual images, even within a category will differ on a panoply of image properties. And since, as the authors note, infants are simply being trained to “find a patch” it is challenging to say what visual properties they are using. A reader might be inclined to accept that the large background patterns invoke categorization, but the targets themselves are quite small (providing less ‘evidence’ for a category) and embedded in a cluttered background. Why would we think this is ‘sufficient’ to trigger categorization? The paper would be strengthened by a more deliberate, rationalized explanation of the various “visual properties”. Why were these attributes measured in the first place? Why these attributes and not others? What are the units? What are the ranges for the images used? Are they meant to be exhaustive, i.e. if target detection can’t be attributed to one or more of these differences, are we to be convinced that the only reasonable conclusion is that detection is due to category membership? Can we see some side-by-side accounting of within vs. between category targets in terms of all the low level visual properties the authors measure (or even additional ones related to fourier spectra)? If we rank order the test stimuli by RT and/or success, does a pattern emerge?

Overall though, I am mostly struggling with an even more general issue. If we accept what seems to be the pattern of results, that purportedly same-category targets facilitate search, doesn’t it then become more parsimonious to think that category is not at play at all here? Somehow I feel like the interpretation is caught in a dilemma. All models of visual search and texture segmentation etc. are based on difference, and promote ‘oddballs’. I do not think the authors can seek to overturn that literature and that principle, and, of course, logically, the target cannot be found unless it has some difference, on some dimension, from the background. So, then, we have to determine what the difference is that infants are picking up on here (and, again, especially so with the purportedly “same category” stimuli). It can’t be category membership, per se, logically, because that produces a lack of a difference in this context (i.e., some category detector, running over a same-category stimulus here would find nothing of interest, just, say, vegetation all around). The only differences I can think of then are 1) heightened sensitivity to category exemplars, that somehow, the infant visual system looks for by default, and notes, e.g., “there is a type of vegetation all around in this image, and here is a spot that’s also vegetation, but a different kind of vegetation”. And, further, that these within-vegetation contrasts are given higher ‘scores’ (data-driven salience, driving search) than between category contrasts (say, an artifact patch on the vegetation background). Or, 2) somehow the set of within-category stimuli used here, unluckily, had a statistically greater contrast along some other low-level, non-category-relevant feature dimension. Do the authors agree with this breakdown? Is there something I’m failing to consider? Then, given that 1) is so counterintuitive, to me, 2) becomes more likely and I think the authors need to do some more work to rule it out. Some kind of targeted replication, plus a deeper dive into the specifics of these images would help (as noted in my point above). Are there other aspects of the data/analyses the authors could provide post-hoc to corroborate their interpretation?

6. PLOS authors have the option to publish the peer review history of their article (what does this mean?). If published, this will include your full peer review and any attached files.

Reviewer #1: No

Reviewer #2: No

---

## [Author Response · Author response to Decision Letter 0]

23 Dec 2021

Reviewer #1: There is increasing interest in how we develop sensitivity to natural scene statistics, and how infants learn to segment complex visual scenes and attend to relevent information, and this study has a significant contribution to make to this area of research. The task is a visual search task where infants are presented a target circle on a background - the 'category' that the target is from is either a congruent natural texture, or incongruent. The authors conclude that performance on this task is a result of a combination of perceptual and categorical properties of the stimuli.  

The paper represents a huge amout of work, both in image and data analysis and in data collection all of a high quality. There are some areas that should be addressed, either in a response to review or making edits to the manuscript. We have chosen major revisions only because it seems there is some work to be done to make the authors argument clearer, and does not reflect the quality of the study itself. We're looking forward to reading the final published paper in a journal club in the not too distant future. *note, paper reviewed with the assistance of doctoral researcher, hence 'we' throughout!   

Response:

We thank the reviewers for their positive assessment of our work. We will place our responses underneath each of the comments in italic style. Additionally, we have numbered the comments (e.g., R 2.1, R 2.2, etc.) and will use these numbers to refer to a specific comment. 

- Infant perception of stimuli: Depth, Pixel-wise measures, and use of colour stimuli

R. 1.1

1. The use of depth congurency is an interesting measure. As the authors note, infants this age are sensitive to some depth cues. However, there is evidence that children don't necessarily combine cues to achieve adult like depth perception. Does this have implications for the relevance of the congruency of depth cues in this study? As the authors themselves note in the discussion, it's hard to disentangle 'depth' from images with higher perceived depth having more high contrast areas which are likely to attract the infant.

Nardini, M., Bedford, R., & Mareschal, D. (2010). Fusion of visual cues is not mandatory in children. Proceedings of the National Academy of Sciences, 107(39), 17041-17046.

Thank you for this helpful comment. Our decision to include depth congruency as a covariate along with category congruency was based on a previous study using images from the same image-set used in the current manuscript (Schlegelmilch & Wertz, 2020, PsyArXiv). In the previous study, pictorial depth strongly predicted classification and similarity judgments in preschool-aged children. Therefore, in order to prevent biases when analyzing the effect of category membership on infants' search performance, we included depth congruency as a covariate to balance depth cues with category membership. We state this in the manuscript on p. 14, line 303. As we noted in the discussion (p. 33, line 684) we agree with you that for the effect of the depth rating of the background image, it is difficult to disentangle depth from lighting or contrast cues. This needs further investigation. In order to increase interpretability of some of the infant results, in particular those for category information and rated background properties, we have now added a comparison with adult data from a pilot study in which participants performed the same task (see p.11, line 224 and section "Participants, p.12, line 236). Adults' performance was affected by depth-congruency, and not significantly hindered by high background depth. Therefore, the adult data did not resolve the open question about the particular significance of two-dimensional depth on infants' attention, but they do demonstrate that both factors impact performance across the lifespan.

Thank you also for the reference. It certainly addresses the integration of visual properties during explorative behavior and we have added it to the manuscript (see p. 33, line 684, reference 136). However, the findings of Nardini et al. (2010) do not directly relate to our investigation, because they added three-dimensional depth (i.e., stereoscopy: disparity between the eyes) as second visual property, whereas in the case of pictorial depth-cues in photographs, several two-dimensional cues occur (e.g., shading, contour junctions). Parallel-occurring pictorial depth cues do not change with the maturation of the organism, as Nardini et al. claim for stereoscopy, but may underlie perceptual learning. However, combinations of different two-dimensional cues were learned and increased saliency of depth in other studies using graphic stimuli (for reviews see manuscript reference: Kavšek et al., 2012).

R 1.2

2. Many of the measures used by the authors are based on pixel wise measures (e.g. mean luminance calcualted with each pixel), and some of the differences in stimuli levels will have likely been small. Are these realistically discriminable to an infant, and can any of the findings be explained by a limited infant visual system?

Thank you for pointing this out. The question you raise is very important in the context of the current investigation. As we outline in the introduction, visual abilities are still maturing up into adolescence and fine variations in contrast and small details are very likely still difficult for infants of this age (see p.4, line 89). It is therefore an important finding of the study, that even though statistical properties included fine-grained visual information, infants still reacted to variability in these properties. This is a point we stress in the discussion (see p31, lines 644 ff).

It is possible that naturalistic stimuli affect gaze differently than artificial stimuli because the visual system is geared towards solving visual tasks presented by real-world environments. We made the importance of experience and ecological significance on sensitivity to visual regularities now more central in the introduction (p. 5, lines 102 ff), see also the section "Limitation and future questions" (p. 41).

R 1.3

3. The study uses monchromatic stimuli in 3 colours R,G, and B to try and encourage infant engagment with the task. This unfortunately may have inadvertantly added noise to the measures the authors are collecting data from. For example, brightness perception varies as a function of hue - two colours of equal luminance do not necessarily appear equally bright (Helmholtz–Kohlrausch effect). Although the authors don't find a main effect of colour on performance, which does reassure somewhat that there's no 'hidden bias' brought in by using colour in this way, the authors should be aware that values calculated on greyscale versions of stimuli may not neccessarily reflect perception of chromatic stimuli.

Thank you for raising this issue. We were aware that alternating the background color of the stimuli might introduce some noise, but at the same time, previous work has shown that well-controlled changes in background color keeps infants engaged in a task over many trials and increases eye-tracking data quality (Schlegelmilch & Wertz, 2019, Infancy). Therefore, given the number of trials needed for this task, we decided to make use of alternating stimuli colors while taking steps to ensure that they did not interfere with the experimental conditions. 

When transforming the stimulus color, we took care that perceived brightness did not vary strongly between the colors by (a) reducing saturation in the HSL coordinates, thereby decreasing the Helmholtz–Kohlrausch effect, and (b) by using hues of identical distance to each other which were distributed between the pure CMYG or RGB colors (red, yellow, green, cyan, blue, magenta). The pure colors differ more strongly to each other in brightness than the in-between hues. 

Importantly, during the color transformation, we did not further adapt luminance levels, because this might have reduced data quality due to changes in pupil size.

To address the concern that monochromatic colors alternating between trials increased noise in the data, we assessed the effect of these alternations on performance (see section B in S1 text). This showed that infants' performance in trials preceded by an alternating color did not differ from trials with the same color as the previous trial. This was true for the success-hit rate (Mchange = .37, Msame = .36, t = .2, p = .85), and for latency (Mchange = 1519 ms, Msame = 1616 ms, t = -.8, p = .44). See also response to comment R 2.12

R 1.4

- Categorisation

3. Categories: the paper leans heavily on the use of the word 'category' when discussing the rationale and findings. The paper states that 'infants attend to combinatons of category AND property related cues to distinguish naturalistic patterns' implying that the paper considers categories to be an entirely separate entity to the lower level property related cues. We think that the main evidence for there being a category effect is most effectively shown in figure 3 - by there being a greater chance of target detection for congruent stimuli when contrast differences are large, and the inverse effect when contrast differences are smaller. Are there alternative explantions that don't call on categories - for example is this is actually just an ability to spot an outlier in a statistical distribution rather than an effect of categories?

Thank you for this question. Similar concerns were raised by Reviewer 2 (see R 2.13 to R 2.16).

The (possible) effect of category membership on scene segmentation was central in the previous version of the manuscript because the question of how infants respond to certain superordinate categories led to the current investigation. Indeed, as you point out, our results show that the impact of category information on detection success depended on the low-level salience of the target (i.e., the variable diff_luminance). We now added comparisons of property levels within the factor category-congruency, to rule out biases due to our selection of properties (Table 5).

Moreover, we are in complete agreement that it is difficult to separate visual properties from category information. Accordingly, we do not see categories as independent or opposite from visual properties. Categories must necessarily be defined by a set of properties as we stated now in the discussion (p. 36, line 769). Additionally, some visual properties might receive particular attention in infants, implying a certain significance of their own. Therefore, in the revision of the introduction and discussion we have now leaned less on category membership, but discussed alternative reasons for facilitating effects on detection success and latency. We put more weight on visual information that is of significance to humans (and particularly to infants) that can relate to category AND property (see sections: "The significance of category information", " Depth cues, but not shape predicted detection performance", " Infants' visual search relates to preschoolers', but not adults' similarity judgments", "Conclusion").These changes describe an interdependency between category and visual property, but emphasize that there is visual information that can be relevant to vision development because it is of significance in the age-related tasks and /or has been so over evolutionary time. 

However, we still think it is important to consider differences in the visual processing hierarchy between statistical properties and general categories, and related differences in the complexity of neural computations (see p. 34, line 728). In addition, the inclusion of the comparison sample of adults in the revised manuscript added in response to one of Reviewer 2's comments below may provide some further insight into the relationship between categorization development and sensitivity to visual properties (see section on how infants' visual search relates to preschoolers', but not adults' similarity judgments, p.34). 

R 1.5

- Clarity of paper and general comments

4. The paper makes a lot of effort to be clear - the table of definitions is very good and helpful, and overall, the writing is excellent. However, as a result of there being so much included in the paper, it does mean it is in places confusing, or that the key findings get lost in the paper. We're not advocating that the authors remove sections from the paper, but we do think the paper might benefit from a heavy edit for consicesness. The paper has a huge amount to offer which is currently being lost a little along the way.

Thank you for the positive feedback on our writing. We agree with your assessment of the previous version of the manuscript and have therefore edited the revised version with a careful eye towards concisely conveying the key arguments and points. However, addressing some of Reviewer 2's comments required including more content into the main text. The overall length of the revised manuscript is similar to the original submission, but with the many changes we made, we hope we have succeeded in improving the clarity and conciseness of the text.

R 1.6

5. Should 'intensity' be contrast or luminance throughout? All the variables listed could be measured in 'intensity' so it was a little confusing in places.

Thank you for pointing this out. We took the term "intensity" from the literature, including the infant literature (see e.g., Itti and Koch, 2001; Kwon et al., 2016). However, we agree that the term can be confusing and now refer to intensity as "luminance" throughout the manuscript. 

R 1.7 

line 347 - one of these 'incongruents' should be 'congruent' - or alternatively we have misunderstood the way that the model fits to the data.

The sentence was correct, but we agree that it was formulated in a confusing way. We have now edited it. 

Reviewer #2: 

This paper investigates infants’ ability to recognize and discriminate visual patterns by virtue of their category (vegetation, artifact, non-living natural) membership. This was assessed through a visual search task, where a small target patch of an image was embedded in a background image. The target patch was always drawn from a different image, but that image could be from the same category as the background, or from a different category. There are various detours and other considerations, but the overarching hypothesis is that category membership, per se (as opposed to various concomitant low-level visual differences that manifest between images from different categories), would be noted and drive looking toward the target. In general, the work is sound, I really appreciate this area of investigation, and the melding of natural scene image analysis and psychophysics in an infant study. It is a nice niche that would benefit from more work. That said, there were aspects of the study (and the interpretation of results) where I had some concerns.

Thank you for your positive assessment of the general soundness and contribution of our work.

Overall, the exposition itself, especially around methods and results, sometimes lacked clarity and motivation, and could be more refined and deliberate. I will try to offer some concrete suggestions here.

R 2.1

I had concerns with the data screening. As it stands, the screening is based on behavioral outcomes (throwing away a “hit” because recorded gaze was <80%, but applying a different criteria for miss trials). This seems potentially problematic. I would strongly encourage the authors to apply just one, erring-on-the-side-of-inclusivity, criteria across the board, before any considerations of performance or outcomes.

Thank you for mentioning this concern. In response to your comment, we performed an exploratory analysis using similar criteria for hits and misses, but this approach led to the exclusion of even more data (see below for details). Therefore, we would like to explain the motivation for the inclusion criteria we used. 

We originally decided to apply inclusion criteria based on proportion of recorded gaze, because low data quality caused by movement including look-aways can dramatically change the results (e.g., Hessels et al., 2015; Schlegelmilch & Wertz, 2019), particularly when using areas of interest (AOIs; Holmqvist et al., 2012). However, we saw the necessity to calculate the recorded proportions differently between hits and misses. The more conservative inclusion of 80% of recorded gaze in hit trials reduces noise in the latency variable (i.e., long periods without contact to the eyes that occurred for unknown reasons). It also reduces false positives resulting from low data quality. If the same criterion would have been applied to misses as it was to hits, infants would have needed to attend to the screen for at least 3600 ms (80% of the 4500 ms trial length). This would have heavily affected success-rate: In the raw data, as well as in the data used for the original analysis, the success-rates are approximately identical with .37. In contrast, when applying the .8 criterion to both hits and misses, the success-rate is .52. Thus, using the same criterion that we had used for hits in the original analysis also for misses artificially inflates the success rate. 

We decided to apply a data-driven alternative criterion for misses, because a data driven approach is sufficiently neutral and avoids such issues. We used the median of latency in all hit trials (1240 ms) as the necessary minimum of search duration in misses. This led to the reported number of trials (hit: N=459, miss: N = 758; compared to the raw data with hit: N = 500, miss: N = 837). 

In response to your comment, we compared the effect of different thresholds on the proportions of hit and miss trials. We hoped to find a threshold that leads to a similar proportion of hits and misses in the data and sufficiently reduces noise. A threshold of 70% led to hit: N = 472, miss: N = 496. Yet, with this criterion, 249 trials more than in the current analysis would need to be discarded. 

Thus, we still prefer the criteria applied to the data in the original manuscript version. We added a justification of the inclusion criteria in the section on preliminary analysis and data reduction ( p. 18, line 393). 

Nevertheless, we conducted a *preliminary* re-analysis of the main models using the criterion of 70%, see the SI document for this response file, Review-SI Text. The preliminary results of this analysis show that, overall, the pattern of results is similar to the analysis included in the main text of the manuscript. For success (Review Tables 1-3), the effect of category congruency still depends on differences in luminance. Yet, in the models analyzing visual properties, the statistical properties become more influential, while in rated properties, the former significant effect of depth on detection success became marginal. These changes possibly result from the reduction of noise in misses due to the stricter threshold, but do not alter the conclusions in the manuscript.

For the analysis of latency (Review Tables 4-6), the effects of visual properties remained similar to those reported in the original manuscript version. 

R 2.2

There were phrases scattered throughout the text that had the feeling of technical terms, but had vague and unclear meaning, such as “physically intense cues”, “perceptual difficulty”, “prominence”, “familiarity”, “level of property”, “property value”, “less [/more] distinct category combinations”, “processing advantages”, “discriminated statistically”, “difficulty of the images”. It would help the exposition if these terms were replaced with more specific, definitive ones, or at least defined/operationalized.

Thank you for bringing this to our attention. We have now replaced most of the terms and, when that was not possible, we reformulated the sentences to make our meaning clear. 

R 2.3

In places, the technical terms themselves could be sharpened. Why not just call “intensity” / “low-level intensity” / “physically intense cues”, simply mean luminance? Why not call “diff_mean” ‘diff_scaleInvariance’? Etc.

Thank you for this comment. We agree that some of the variable names could lead to confusion and changed them in the revised manuscript (see also comment R 1.6). Specifically, as suggested, we now refer to "intensity" as "luminance" and "diff_luminance", we renamed the variable "area" to "deviation" to reduce misunderstandings, and "child_similarity" to "child_dissimilarity" to be consistent with the direction of the variable.

R 2.4 

Sometimes this can affect understanding of central claims. For instance, I am not clear what is meant by “...visual property could influence infants' search performance in two non-exclusive ways: a) their prominence within a background image might hinder the detection of the target.” Here, it is not clear (to me) what is meant by “prominence” of dimensions that have no natural valence? Could the authors reword and clarify?

Thank you for pointing this out. We have now changed the wording of this sentence to make it more clear (p. 10, line 213).

R 2.5

I think the authors can make a stronger case for “Were targets detected by coincidence” I would be interested to see other comparisons between the target aoi, and the average of the other 9 aoi’s, e.g.: # fixations until aoi (i.e. target aoi vs. average of other 9 aois), time to aoi, dwell time on aoi, ‘success rate’ (proportion of trials on which target, versus other 9 aois, was reached). These “chance levels” (‘coincidence’) should be reported wherever possible (e.g. Figure 1 and Figure 3) since they give a good frame of reference, at least for Intensity 0 conditions.

Thank you for these suggestions. 

When evaluating the coincidence of target detection, we agree that the comparison between fixations to targets and to each of the 9 non-target AOIs, respectively, can serve as a conservative and precise measures. Following your suggestions, we have now added (a) comparisons of 1st fixations on the target compared to the mean N of first fixations on the 9 non-target AOIs (see p.28, lines 582 ff), and (b) a plot showing the proportion of trials in which the first AOI fixated was the target relative to all the trials in which first fixations landed on target and non-target AOIs (see Fig 6). The plot includes the chance level. Please note that infants' fixations frequently landed on stimulus locations that were not possible target locations (i.e., the background image around the circles of possible target locations). The rate of first fixations on targets relative to any first fixations to the stimulus is given in Table 2. In addition, the frequencies of the appearance of a target on one of the AOIs in the actual trials, the success rate related to these frequencies by location, and the rate of first fixations are reported in the ----SI, S1 Table. 

Because the trial always ended after target detection (see also R 2.9), dwell time cannot be compared between target and non-target AOIs.

R 2.6

I was a little unclear on how many trials and subjects contributed to each ‘data point’ (e.g. target-background combination, or at least categories of target-background). Something about the math was not clear to me (“27 images on 10 possible locations and presented in three different colors led to 260 different stimuli”). More detail could be given about the data itself, and the breakdown by conditions, colors. If my math is right, it works out to be about 5 trials per image, 15 per image if we collapse over color? But, those are divided by 4 if we wanted to, say, just compare performance at intensity 0?

Combining the 27 images with each other as targets and background led to 729 possible image combinations. However, these needed to be reduced since we only included combinations that were (a) balanced over categories, (b) were congruent or incongruent in category-combination and their levels of depth, (c) and neither target nor background images repeated more than twice in each version of the experiment. This led to 261 different stimuli, within which we chose moderately salient target locations (this information is included in the Stimuli section in the Methods, and the caption of Fig 1). These criteria were difficult to meet, which is the reason that not all images are included in equal number. The frequencies of the defining factors in the stimuli of the eight experiment-versions (categories of targets and backgrounds, color, location, etc.) are now added in the SI S1 Table.

Concerning your question about "intensity 0", please refer to the answer to comment R 2.13.

R 2.7

I think figure 1 has an incorrect y axis on the scatter plots (I expect it to be RT in ms) or am I missing something?

The y-axis on this figure was correct. The y-axis referred to property level, and the x-axis to latency or success. However, following your suggestions below concerning data visualization (R 2.11), we have now exchanged this figure with two new figures that we believe are less likely to be confusing (Fig 4, Fig 5).

R 2.8

I do not understand how the authors are using hits and misses when, typically defined, misses are just 1-hits. Why not just code performance as percent correct?

There were several reasons why we chose the current comparison between hits and misses: (a) this way, we could directly compare the stimuli properties between miss and hit trials; (b) it would have been difficult to find parameters for which to assess "percent correct", because infants viewed different extracts of the 261 stimuli; (c) the separate analysis of single miss and hit trials allowed us to include random intercepts beyond id (i.e., background image, target location) which accounted for variability that would have been hidden in more global percentages.

R 2.9

Apologies if I missed it, but what latency is entered if the target is not found (miss)?

Only trials in which a target was detected were included in the latency analysis. This is stated on page 19, line 430.

R 2.10

Nearly everything - certainly all the figures - from the “supplementary materials” need to be in the main text. As well, the figures could use more annotation and labels, and more detailed captions.

Thank you for these suggestions. Following your advice, we have now included some of the supplementary figures in the main manuscript (S1 Fig, S2 Fig), and also some of the supplementary texts (S2 text A, B). We also added new figures (see comment R 2.11). A few of the figures have remained in the supplementary information in order to keep the main text as concise as possible.

R 2.11

The statistics wind up being a bit complex due to all the factors and varying tests in different contexts. I think the paper could do with some more data visualization. (As it is, we only have Figure 2, which does not even have data points, and the caption does not say anything about the nature of the fits, etc.). Some of this would be mitigated by my earlier suggestion to move other Figures and information from the Supplement into the main text.

Thank you for this suggestion. We now added two figures (Fig 4, Fig 5) that also give confidence intervals of the marginal effects.

R 2.12

Do the authors have a reference for the alternating-color-stimuli design they used? This seems a rather extreme way to help maintain vigilance. I know the authors found that color did not predict performance, but what of differences between the colors?

Yes, we do have a reference for this. We investigated the effect of alternating colors in a previous study on eye-tracking data quality: Schlegelmilch, K., & Wertz, A. E. (2019). The effects of calibration target, screen location, and movement type on infant eye-tracking data quality. Infancy, 24(4), 636–662. This reference is cited in the manuscript to justify this aspect of our experimental design (see p.15, line 337).

There were no differences in performance between the colors ( see p. 20, line 453). We also investigated the effect of alternating colors in response to comment to R 1.3 and found that it did not affect target detection either (see Section B in S1 text).

R 2.13

Now, turning to the results themselves and their interpretation. For what I think is the most natural, straightforward comparison - the ‘success rate’ of whether infants found target patches differentially, depending on if it had a category match/mismatch with the background - they seem to have a null result. This is captured in Figure 3 at intensity level 0 (intensity refers to mean luminance, which, in levels 1-3, was artificially increased on target patches to facilitate infants’ search). In this comparison, there is no difference between success as a result of target-category membership - the core contrast of the whole study - and actually, target direction rates themselves are so low as to approach chance (i.e. fixating the target ‘accidentally’ as the infant simply scans the scene). Null results are fine of course, but the introduction and discussion (layering in further analyses and speculation) tended to bury the lede here. Do the authors agree? Shouldn’t this finding be more central in the discussion?

Yes, we agree that the effect of category information on detection performance only occurs in interaction with differences in luminance. In the revised version of the manuscript, we are discussing the result accordingly. 

Please see responses to:

- the result of category-congruency below in R 2.16.

- your concerns on the effect of luminance in R 2.14 

R 2.14

Then, why was mean luminance (intensity) varied at all? This manipulation needs a lot more justification and explanation, especially as it winds up being the central driver of the “main” (counterintuitive and unexpected) results, given its interaction with category membership. It is hard to think of any reason to expect luminance to interact with categorization. In fact, I think the default stance would be that putting category information on top of a (much more impactful) cue like relative luminance would tend to either effectively discourage the use of category information (by rendering it largely unnecessary), or, as a practical, ‘signal/noise’ matter, tend to obscure any relatively small effects of category against the backdrop of much larger effects of luminance.

It is important to note that differences in luminance were NOT artificially increased or decreased. Instead, luminance (precisely, diff_luminance) was assessed from the target-background image combinations similar to how the other statistical properties were assessed (see p.14, line 309). Like these other properties, it is a continuous variable. 

It is not clear to us which part of the text you referred to, when assuming the manipulation of an artificial 4-stage increase of 0 luminance. We hope that the current version of the manuscript does not lead to the same misunderstanding. 

Luminance is one early attention-grabbing property (see introduction p.4 line 90; p.9 line 189). Due to its strong predictive value on infants' gaze described in the developmental literature (e.g., Frank et al., 2014; Kwon et al., 2016; Sireteanu et al., 2005, see main references), it was included in all our models as a control variable with the intention of separating the predictors' variance in the data related to luminance contrasts from variance related to structure or content. We have now added further information on the inclusion of the covariate diff_luminance in the revised manuscript (see also Table 1 p. 10; results, p.21, line 462, discussion p.30, line 633) 

We agree that it is important to understand if greater differences in luminance increased the probability to detect otherwise less salient visual information. A significant improvement of the model's fit when adding the interaction term between diff_luminance and the other predictors indicated such a supporting effect of diff_luminance with category-congruency. We clearly stated in the revised discussion that category-congruency only affected detection success in combination with luminance (see results p.30, line 617, p.36, line 754, and discussion p. 36, line 761)

R 2.15

Then, while I am sympathetic to the author’s attempts to link the present results to category formation, I am hoping the authors can make a stronger case (this point is relevant also to my next one below). All the individual images, even within a category will differ on a panoply of image properties. And since, as the authors note, infants are simply being trained to “find a patch” it is challenging to say what visual properties they are using. A reader might be inclined to accept that the large background patterns invoke categorization, but the targets themselves are quite small (providing less ‘evidence’ for a category) and embedded in a cluttered background. Why would we think this is ‘sufficient’ to trigger categorization? The paper would be strengthened by a more deliberate, rationalized explanation of the various “visual properties”. Why were these attributes measured in the first place? Why these attributes and not others? What are the units? What are the ranges for the images used? Are they meant to be exhaustive, i.e. if target detection can’t be attributed to one or more of these differences, are we to be convinced that the only reasonable conclusion is that detection is due to category membership? Can we see some side-by-side accounting of within vs. between category targets in terms of all the low level visual properties the authors measure (or even additional ones related to fourier spectra)? If we rank order the test stimuli by RT and/or success, does a pattern emerge?

Thank you for this comment. We agree that the paper benefits from more explanation of the visual properties we tested in this paper. Because it is impossible to assess an exhaustive amount of visual properties that might underlie visual categorization (see also Limitation section), we analyzed a selection of visual properties. We chose properties that we thought might be relevant for image segregation due to their potential to distinguish the categories used our image set, and due to findings from previous research indicating a role in visual categorization in adults, or in infant visual development (see p.5, line 109 and p.8 line 177 ff). 

Following your suggestion, we have now added more information about our rationale for selecting these visual properties (Table 1). We also made the data of the 261 stimuli's properties available on osf: https://osf.io/uyg76/?view_only=14e8e992abfe46e992e5a963776fc70b and added Table 5 describing properties as function of category-congruency for all trials included in the eight experiment versions (p.37).

However, we also like to mention that although the target patches are small, infants are clearly able to distinguish them from the background. We have confirmed that this is not due chance. Therefore, infants must be representing the small target patch as sufficiently different from the background based on some combinations of properties. Our goal was to examine some properties, including categorical information, that might underlie this ability.

R 2.16

Overall though, I am mostly struggling with an even more general issue. If we accept what seems to be the pattern of results, that purportedly same-category targets facilitate search, doesn’t it then become more parsimonious to think that category is not at play at all here? Somehow I feel like the interpretation is caught in a dilemma. All models of visual search and texture segmentation etc. are based on difference, and promote ‘oddballs’. I do not think the authors can seek to overturn that literature and that principle, and, of course, logically, the target cannot be found unless it has some difference, on some dimension, from the background. So, then, we have to determine what the difference is that infants are picking up on here (and, again, especially so with the purportedly “same category” stimuli). It can’t be category membership, per se, logically, because that produces a lack of a difference in this context (i.e., some category detector, running over a same-category stimulus here would find nothing of interest, just, say, vegetation all around). The only differences I can think of then are 1) heightened sensitivity to category exemplars, that somehow, the infant visual system looks for by default, and notes, e.g., “there is a type of vegetation all around in this image, and here is a spot that’s also vegetation, but a different kind of vegetation”. And, further, that these within-vegetation contrasts are given higher ‘scores’ (data-driven salience, driving search) than between category contrasts (say, an artifact patch on the vegetation background). Or, 2) somehow the set of within-category stimuli used here, unluckily, had a statistically greater contrast along some other low-level, non-category-relevant feature dimension. Do the authors agree with this breakdown? Is there something I’m failing to consider? Then, given that 1) is so counterintuitive, to me, 2) becomes more likely and I think the authors need to do some more work to rule it out. Some kind of targeted replication, plus a deeper dive into the specifics of these images would help (as noted in my point above). Are there other aspects of the data/analyses the authors could provide post-hoc to corroborate their interpretation?

(Please also see response to reviewer 1's comment R 1.4 where similar concerns were addressed.)

Your comment R 2.16 is raising fundamental concerns about the finding on category congruency. It is therefore important to note that in the current study, the investigation of visual properties affecting detection performance--independent of category membership--was as important to us as that of the general categories. We indicated this interest in the revised introduction now more clearly (p.4 line 76).

When designing the experiment, we took care that the category-combinations were approximately balanced in the selected visual properties (see Table 5 and the description of depth-congruency (p. 14, line 303). Therefore, we do not think that the within-category stimuli inadvertently had greater contrast along those properties. However, we cannot rule some properties that we did not asses might have led to a bias in category-congruency. We added this point to the Limitations section. We also stated clearly, that congruent categories only led to a higher probability to detect a target if combined with greater diff_luminance, whereas incongruent categories led to better detection performance than congruent categories if the full range of diff_luminance is taken into account, and they were affected less by differences in luminance (p.22, line 489). 

We also fully agree with you that further investigations are needed. However, it would inflate the probability to find false positives if we added more visual properties in the current investigation, and ran more models on the data (see e.g. Simmons et al., 2011). Thus, following your concerns and our continuing wish to understand the result on category congruency, we added an adult sample performing the same experiment in a recent pilot study (see p. 12, line 236, and section on preliminary analysis and data reduction p. 18). In adults, luminance differences did not interact with category-congruency, and there was no main effect of category-congruency (p.24 line 519, Table 3 and Fig 4), suggesting that adults were less affected by the luminance differences and their combination with category information in our stimuli than infants. In the Limitation section we now suggest future investigations with different stimuli: Target-background combinations of different superordinate categories could be compared to combinations of sub-groups within particular superordinate categories, p. 39 line 832. 

In the revised version of the manuscript, we now substantially reworked parts of the introduction to make our view on the interrelation between visual properties and category membership more clear (sections on the significance of category information, and on the current investigation). We also discussed the interaction between category-congruency and differences in luminance in a more general way without addressing differences between the factor-levels (i.e., congruent vs. incongruent combinations; see discussion, section "Did categorical information affect infants' detection performance?" p. 36). 

In closing, we would like to thank you once again for your thorough and challenging feedback. We hope that the changes we have made to the manuscript sufficiently answer your questions and dispel your concerns.

 References:

Frank, M. C., Amso, D., & Johnson, S. P. (2014). Visual search and attention to faces during early infancy. Journal of Experimental Child Psychology, 118, 13–26.

Kwon, M.-K., Setoodehnia, M., Baek, J., Luck, S. J., & Oakes, L. M. (2016). The development of visual search in infancy: Attention to faces versus salience. Developmental Psychology, 52(4), 537–555. 

Simmons, J. P., Nelson, L. D., & Simonsohn, U. (2011). False-positive psychology: Undisclosed flexibility in data collection and analysis allows presenting anything as significant. Psychological science, 22(11), 1359-1366.

Sireteanu, R., Encke, I., & Bachert, I. (2005). Saliency and context play a role in infants’ texture segmentation. Vision Research, 45(16), 2161–2176.

---

## [Decision Letter · Decision Letter 1]

16 Mar 2022

Visual segmentation of complex naturalistic structures in an infant eye-tracking search task

PONE-D-21-01158R1

Dear Dr. Schlegelmilch,

We’re pleased to inform you that your manuscript has been judged scientifically suitable for publication and will be formally accepted for publication once it meets all outstanding technical requirements.

Kind regards,

Guido Maiello

Academic Editor

PLOS ONE

Additional Editor Comments (optional):

Reviewers' comments:

Reviewer's Responses to Questions

**Comments to the Author**

1. If the authors have adequately addressed your comments raised in a previous round of review and you feel that this manuscript is now acceptable for publication, you may indicate that here to bypass the “Comments to the Author” section, enter your conflict of interest statement in the “Confidential to Editor” section, and submit your "Accept" recommendation.

Reviewer #2: All comments have been addressed

2. Is the manuscript technically sound, and do the data support the conclusions?

Reviewer #2: Yes

3. Has the statistical analysis been performed appropriately and rigorously? 

Reviewer #2: Yes

4. Have the authors made all data underlying the findings in their manuscript fully available?

Reviewer #2: Yes

5. Is the manuscript presented in an intelligible fashion and written in standard English?

Reviewer #2: Yes

6. Review Comments to the Author

Reviewer #2: Thank you for your thorough response to my comments. I think the paper is clearer and stronger now, and I appreciate the work that went into the revisions!

7. PLOS authors have the option to publish the peer review history of their article (what does this mean?). If published, this will include your full peer review and any attached files.

Reviewer #2: No

---

## [Editor Report · Acceptance letter]

24 Mar 2022

PONE-D-21-01158R1 

Visual segmentation of complex naturalistic structures in an
infant eye-tracking search task 

Dear Dr. Schlegelmilch:

I'm pleased to inform you that your manuscript has been deemed suitable for publication in PLOS ONE. Congratulations! Your manuscript is now with our production department. 

Kind regards, 

on behalf of

Dr. Guido Maiello 

Academic Editor

PLOS ONE